# A Systematic Genotoxicity Assessment of a Suite of Metal Oxide Nanoparticles Reveals Their DNA Damaging and Clastogenic Potential

**DOI:** 10.3390/nano14090743

**Published:** 2024-04-24

**Authors:** Silvia Aidee Solorio-Rodriguez, Dongmei Wu, Andrey Boyadzhiev, Callum Christ, Andrew Williams, Sabina Halappanavar

**Affiliations:** 1Environmental Health Science and Research Bureau, Health Canada, Ottawa, ON K1A0K9, Canada; silvia.solorio.rodriguez@gmail.com (S.A.S.-R.); dongmei.wu@hc-sc.gc.ca (D.W.); andrey.boyadzhiev@hc-sc.gc.ca (A.B.); callum.christ@hc-sc.gc.ca (C.C.); andrew.williams@hc-sc.gc.ca (A.W.); 2Department of Biology, University of Ottawa, Ottawa, ON K1N6N5, Canada

**Keywords:** soluble nanomaterials, poorly soluble nanomaterials, metal oxide nanoparticles, alveolar epithelial cells, micronucleus, genotoxicity, comet assay, lung toxicity

## Abstract

Metal oxide nanoparticles (MONP/s) induce DNA damage, which is influenced by their physicochemical properties. In this study, the high-throughput CometChip and micronucleus (MicroFlow) assays were used to investigate DNA and chromosomal damage in mouse lung epithelial cells induced by nano and bulk sizes of zinc oxide, copper oxide, manganese oxide, nickel oxide, aluminum oxide, cerium oxide, titanium dioxide, and iron oxide. Ionic forms of MONPs were also included. The study evaluated the impact of solubility, surface coating, and particle size on response. Correlation analysis showed that solubility in the cell culture medium was positively associated with response in both assays, with the nano form showing the same or higher response than larger particles. A subtle reduction in DNA damage response was observed post-exposure to some surface-coated MONPs. The observed difference in genotoxicity highlighted the mechanistic differences in the MONP-induced response, possibly influenced by both particle stability and chemical composition. The results highlight that combinations of properties influence response to MONPs and that solubility alone, while playing an important role, is not enough to explain the observed toxicity. The results have implications on the potential application of read-across strategies in support of human health risk assessment of MONPs.

## 1. Introduction

Engineered Nanomaterials (ENMs) are materials that are manufactured at or within the nanoscale (1 to 100 nanometers inclusive) or have internal or surface structures in the nanoscale [1]. The internal and external structures are expected to break down into component nanomaterials [2]. Metal oxide nanoparticles (MONPs) are among the most synthesized and utilized ENMs globally, with a wide variety of applications, including in consumer products, electronics, energy, food, agriculture, and medicine sectors [3], with an approximate worldwide production from less than 10 tons to over 10,000 tons per year [4]. This increase in synthesis and application has led to enhanced risks of human exposure to these materials, potentially resulting in adverse health effects [5,6]. Inhalation is the main route of exposure to MONPs occupationally, where workers may be exposed during material handling (weighing, embedding, cleaning, filtration, etc.), whereas consumers could be exposed via inhalation of powders and sprays containing MONPs [7,8,9]. Because of their small size, once inhaled, they can bypass the upper and lower respiratory clearance mechanisms to reach the alveolar region and potentially induce pulmonary toxicity by various mechanisms [10].

Inhalation of mixtures containing incidental MONPs, such as welding fumes, has been associated with an increased risk of lung disease in humans [11]. A recent exposure monitoring study positively identified chromium oxide and nickel oxide nanoparticles (NiO NPs) in the plasma of stainless-steel welders post-shift [12]. Furthermore, analysis of lung samples taken from welders has shown evidence that MONPs co-localize with macrophages and fibrotic foci in lungs in diseased individuals [13]. These data are complemented by recent studies of humans exposed occupationally or on a volunteer basis to engineered MONPs, indicating the potential risk associated with these materials [14].

Limited epidemiological and biomonitoring studies in recent years have raised concerns about occupational exposure to engineered MONPs. In workers employed in titanium dioxide (TiO_2_) nanoparticles (NPs) manufacturing facilities, increased levels of biomarkers associated with cardiopulmonary and systemic toxicity were found in their blood, which was correlated with workers’ exposure to MONPs [14]. Similarly, researchers involved in the production of metal-containing nanocomposite compounds have been shown to have elevated levels of nucleic acid and protein oxidation in their exhaled breath condensate in comparison with non-exposed control subjects [15]. In addition, human volunteers exposed via inhalation to zinc oxide (ZnO) NPs at 1 mg/m^3^ showed signs of systemic toxicity, including increases in blood neutrophils and acute phase proteins [16]. In many countries, such as the US [17], occupational exposure limits to ZnO fume are set at 5 mg/m^3^, which indicates that current occupational exposure limits for metal oxides (MO) may not be protective enough for the nanoforms (i.e., MONPs). These scarce studies underscore the need to better understand the mechanisms involved in the toxicity of MONPs to better protect workers and consumers. Research using in vivo model systems in past years has helped better define the hazard that MONPs can pose in mammalian organisms. For example, exposure to ZnO NPs, copper oxide (CuO) NPs, NiO NPs, manganese oxide (MnO_2_) NPs, aluminum oxide (Al_2_O_3_) NPs, cerium oxide (CeO_2_) NPs, TiO_2_, and iron (III) oxide (Fe_2_O_3_) NPs induce inflammation, oxidative stress, and tissue injury in different animal models [18,19,20,21,22,23,24,25].

Besides inflammation, MONPs have been reported to induce primary and secondary genotoxicity both in vivo and in vitro [26,27]. Primary genotoxicity is the result of direct interactions of NPs with DNA or cellular proteins involved in mitosis or via the generation of reactive oxygen species (ROS) [28]. Secondary genotoxicity occurs when activated phagocytes produce ROS and oxidative DNA damage during inflammation [28]. Comet and micronucleus (MN) assays are two of the most commonly used techniques to evaluate the genotoxic potential of compounds due to their high sensitivity to detect DNA breaks and chromosome damage, respectively [29]. The comet assay (also known as single-cell gel electrophoresis) has three different versions based on the pH of the buffers used for the electrophoresis. The alkaline version with a pH higher than 13 detects single- and double-strand breaks that result from direct interactions with DNA, alkali labile sites, or transient DNA strand breaks from DNA excision repair [29,30]. The MN assay determines the frequency of MN (also known as Howell-Jolly bodies) that are formed by clastogenesis (chromosomal breaks) or aneugenesis (disruption of the mitotic apparatus) when the chromosomal fragment or chromosome that is not integrated into the nucleus of a daughter cell is enveloped by a nuclear membrane [29,31]. Recent advances in technology have led to the development of commercially available high-throughput versions of these methodologies, which are uniquely suited for the task of in vitro genotoxicity screening.

The genotoxicity of MONPs is influenced by their physicochemical properties, such as chemical composition, size, surface properties, agglomeration/aggregation state, and solubility [32]. However, the knowledge of how MONP properties impact their toxicity is not complete. Because of the high number of ENMs with different physicochemical properties that require testing, in vitro high-throughput screening techniques are desired for the rapid screening of potentially hazardous ENMs [33]. In a previous study, Boyadzhiev et al. [34] used the high-throughput CometChip assay to evaluate the genotoxic potential of three types of MONPs, CuO, ZnO, and TiO_2_ NPs and metal oxide microparticles (MOMP/s), as well as their equivalent metal chloride salts, zinc (II) chloride (ZnCl_2_) and copper (II) chloride (CuCl_2_) in mouse lung epithelial cells (FE1). It was found that ZnO NPs and CuO NPs induced a concentration-dependent genotoxic response, with CuO NPs being more genotoxic than ZnO NPs. For Zn, all forms, including NPs, microparticles (MPs), and dissolved Zn, induced DNA damage, suggesting that the metal ion Zn^2+^ is the main contributor to toxicity [34]. Of the TiO_2_ NP variants tested, uncoated, silica, and silica-alumina-coated TiO_2_ NPs induced a concentration-dependent increase in DNA breaks, which was lower in magnitude compared to that observed after exposure to other MONPs. In another study, the genotoxicity of NPs with different chemical compositions, including silicon dioxide (SiO_2_), ZnO, Fe_2_O_3_, silver (Ag), and CeO_2_ NPs, was evaluated in human lymphoblastoid (TK6) and Chinese hamster ovary (H9T3) cell lines using a high-throughput comet assay. ZnO, Ag, and Fe_2_O_3_ NPs induced a concentration-dependent increase in DNA damage with different genotoxic responses between the two cell lines [35]. The potential genotoxicity of Ag NPs coated with citrate or polyvinylpyrrolidone (PVP) and positively and negatively charged SiO_2_ NPs were evaluated in primary human lymphocytes and B lymphocyte human cells (WIL2-NS) using a high-throughput cytokinesis-block MN assay (a version of MN assay for human lymphocytes) at different concentrations. It was observed that NPs of 10 nm size were more genotoxic than 70 nm, and citrate-coated NPs were more genotoxic than PVP-coated NPs, whereas SiO_2_ did not induce MN [36]. In another study, HBEC3-KT cells (an hTERT-immortalized epithelial cell line) were exposed to several types of NPs (silver [Ag], gold [Au], and platinum [Pt]) of two sizes, 5 or 50 nm. Au NPs of 5 nm caused more DNA damage than Au NPs of 50 nm, whereas Pt NPs of 50 nm induced more damage than the smaller size Pt [37]. From these studies, it can be concluded that several properties of MONPs, including chemical composition, surface coating, and size, play a significant role in this response, in addition to solubility. The use of high-throughput screening tools makes it practical to assess many concentrations and property variants simultaneously, allowing for the elucidation of links between physicochemical properties and genotoxicity.

Using such tools, the objective of the current study was to evaluate the genotoxic potential of a wide panel of MONPs with varying solubility profiles (i.e., ZnO, CuO, MnO_2_, NiO, Al_2_O_3_, CeO_2_, TiO_2_, Fe_2_O_3_ [38]). Different surface coatings were included, in addition to MPs and dissolved metal analogs. Health Canada has identified these MONPs and their coatings (PVP, 3-Aminopropyl triethoxysilane [APTES], stearic acid, and silane) as priority nanomaterials due to their commercialization in Canada [38,39,40].

This study examined DNA strand breaks in FE1 MutaMouse lung epithelial cells via the high-throughput CometChip assay and chromosomal damage by the high-throughput MicroFlow MN assay. The data generated by Boyadzhiev et al. [34] were used to compare coated and uncoated nanoparticles.

## 2. Materials and Methods

### 2.1. MONPs, MOMPs, and Dissolved Metal Analogs

A total of 21 individual uncoated and coated ZnO, CuO, MnO_2_, NiO, Al_2_O_3_, CeO_2_, TiO_2_, and Fe_2_O_3_ MONPs were investigated in this study. The material purity ranged from 97.8 to 99+%, and the primary particle size ranged from 10 to 60 nm. Table 1 lists the catalog numbers (Cat#), the vendor, and the physicochemical characteristics of MONPs as provided by the manufacturer.

The study also included the uncoated bulk analogs of ZnO, CuO, MnO_2_, NiO, Al_2_O_3_, CeO_2_, TiO_2_, and Fe_2_O_3_ (Table 2) and the dissolved metal analogs to evaluate the effects of particle size and solubility on the potential genotoxicity. ZnCl_2_ (Cat#Z0152), CuCl_2_ dihydrate (CuCl_2_·2H_2_O, Cat#C3279), manganese (II) sulfate monohydrate (MnSO_4_·H_2_O, Cat#M7899), nickel (II) chloride hexahydrate (NiCl_2_·6H_2_O, Cat#N6136), aluminum (III) chloride hexahydrate (AlCl_3_·6H_2_O, Cat#A0718) and cerium (III) chloride heptahydrate (CeCl_3_·7H_2_O, Cat#228931) were obtained from Sigma–Aldrich (Oakville, ON, Canada). The physicochemical characterization, the results of cell viability and DNA damage assay (as evaluated by comet assay) for ZnO NPs (Cat#US3580), CuO NPs (Cat#544868), TiO_2_ NIST NPs (referred to as “uncoated TiO_2_ NPs”), ZnO MPs (Cat#US1003M), CuO MPs (Cat#US1140M), TiO_2_ MPs (Cat#US1017M), ZnCl_2_ and CuCl_2_ were previously reported [34] and are included in this study for comparison purposes.

### 2.2. Primary Particle Size Analysis of MONPs by Transmission Electron Microscopy (TEM)

In addition to the characterization details provided by the manufacturer, the primary particle size of MONPs was also analyzed in-house using TEM. In brief, dry MONPs were analyzed by a JEM-2100F Field Emission TEM (JEOL, Peabody, MA, USA). Several images of each MONP dry suspension were obtained, and at least 100 individual NPs were sized using ImageJ 1.53g to determine the length and width of NPs. GraphPad Prism Software 9.5.0 (San Diego, CA, USA) was used to create frequency distribution histograms. Scanning electron micrographs of the MOMPs were taken using a JSM-7500F Field Emission scanning electron microscope (SEM) (JEOL, Peabody, MA, USA). The primary size of the MOMPs was not determined due to their size and aggregate state.

### 2.3. Specific Surface Area (SSA) Measurement for MONPs and MOMPs

The SSA of each MONP and MOMP was experimentally determined using the Brunauer–Emmett–Teller (BET) method with nitrogen adsorption. For this purpose, an Autosorb iQ (Anton Paar Canada Inc., Montreal, QC, Canada) instrument was used. The samples were weighed before degassing and then again after degassing. For MO without an organic coating, the samples were heated at 20 °C/min to 120 °C and held for 12 h. Organically coated MO were heated at 20 °C/min to 85 °C and held for 12 h. Samples were run through a built-in leakage detection method to ensure no moisture or organic volatiles were remaining in the sample before proceeding with surface area analysis. SSA was determined by the multipoint BET method.

### 2.4. MONPs and MOMPs Exposure Suspensions

MONPs and MOMPs stock suspensions were prepared according to Boyadzhiev et al. [34] and Avramescu et al. [38,39]. In brief, materials were weighed on an analytical mass balance (XSR105 Mettler Toledo), suspended in UltraPure^TM^ DNase/RNase-Free distilled water (dH_2_O) (Cat#10977015, Life Technologies, Burlington, ON, Canada) and sonicated in an ice-water bath. All suspensions were sonicated using a Branson Ultrasonics Sonifier^TM^ 450 (Branson Ultrasonics Markham, ON, Canada), equipped with a ½” disruptor horn (with extension for volumes greater than 8 mL) and a removable flat tip. The tip was immersed 1–1.5 inches into the suspension from the surface. Appendix A summarizes the sonication details and the delivered sonication energy for each material suspension in water. An aliquot of the stock suspension was used to characterize the particle suspension, as described in Section 2.5. Serial dilutions were prepared from the stock suspension in DMEM/F12 (Dulbecco’s Modified Eagle Medium/Nutrient Mixture F-12) cell culture medium without phenol red (Cat#21041-025, Life Technologies, Burlington, ON, Canada) to obtain different concentrations for cell exposure. Dilutions were inverted 20 times to mix and used within 15 min of preparation.

Following the exposure suspension preparation, it was observed that a fraction of stearic acid-coated CeO_2_ NPs and stearic acid-coated NiO NPs remained unsuspended, floating at the top of the suspension. The well-suspended fraction was transferred to a new vial, and two aliquots from this vial were used to characterize the suspension as described in Section 2.5 and prepare the serial dilutions in the cell culture medium; the remaining unsuspended fraction was stored at −20 °C to determine the exact concentration of the stock suspension as described in Rahman et al. [42], with a slight modification to the protocol.

In brief, aluminum weighing boats (43 mm, Cat#HS14521A, Sigma–Aldrich, Oakville, ON, Canada) were weighed on an analytical mass balance (XSR105 Mettler Toledo), resulting in Weight_1_. A total of 3 mL of each suspension or 3 mL of dH_2_O (the same dH_2_O used to prepare the suspension) was deposited on the top of each boat. All samples were dried with aluminum boats closed in an oven at 80 °C for 8 h. Once the suspensions were dried, boats were acclimatized at room temperature overnight and weighed on the same balance, resulting in Weight_2_. The exact concentrations were calculated by subtracting Weight_2_ from Weight_1_ and dividing the final mass by the sample volume (3 mL). The weight of the boats with dH_2_O was the same before and after the drying process, which corroborated the purity of the dH_2_O and indicated that 8 h was enough to dry NP suspensions. Two or three independent stock suspensions were dried to determine the exact concentrations. The results for the stock suspension of stearic acid-coated CeO_2_ NPs and stearic acid-coated NiO NPs were 0.67 ± 0.14 mg/mL and 2.96 ± 0.43 mg/mL, respectively. The exact concentrations in the medium of those two NPs are presented in parenthesis in the respective figures.

### 2.5. MONPs Characterization by Dynamic Light Scattering (DLS) and Electrophoretic Light Scattering (ELS)

The hydrodynamic size (HD) and polydispersity index (PDI) were determined by DLS, whereas the surface charge (zeta potential [ZP]) was characterized by ELS [43]. MONPs stock suspensions were diluted to 50 μg/mL in dH_2_O or DMEM/F12 without phenol red with 2% fetal bovine serum (FBS) (Cat#12483-020, Life Technologies, Burlington, ON, Canada), 100 U/mL penicillin G and 100 μg/mL streptomycin (Cat#15140-122, Life Technologies, Burlington, ON, Canada), and 1 ng/mL human epidermal growth factor (EGF) (Cat#PHG0311, Life Technologies, Burlington, ON, Canada). From each dilution, 7–10 individual measurements were conducted on a Zetasizer Nano ZSP (Malvern Panalytical, Westborough, MA, USA).

### 2.6. Cell Culture

Immortalized FE1 cells derived from the transgenic MutaMouse model were used in this study. FE1 cells have the characteristics of type I and II pulmonary alveolar cells and have been used previously to evaluate the genotoxicity and mutagenicity of different chemicals and nanomaterials [44,45,46,47,48,49] and are under validation to be used in a standard in vitro mammalian mutagenicity test [48,50,51]. Cells were maintained in Dulbecco’s Modified Eagle’s Medium Nutrient Mixture: F12 HAM (1:1) cell culture medium (DMEM/F12) (Cat#11320-033, Life Technologies, Burlington, ON, Canada) with 2% FBS, 100 U/mL penicillin G and 100 μg/mL streptomycin, 1 ng/mL EGF at 37 °C in an incubator at 95% humidity and 5% CO_2_. DMEM/F12 without phenol red cell culture medium was used for all exposures under the previously described conditions (Section 2.4).

### 2.7. Cell Viability Assay to Determine the Experimental Conditions for the Comet Assay

Trypan Blue exclusion analysis was used to determine the viability following MO or dissolved metal exposure. Cells were plated at 65,000 cells/cm^2^ in a 6-well plate using DMEM/F12 without phenol red and were incubated for 24 h. The following day, cells were exposed to different particle concentrations for 2 and 4 h at 37 °C with 95% humidity and 5% CO_2_. Each well was washed with 0.5 mL Dulbecco’s phosphate-buffered saline (DPBS) (Cat#14190-144, Life Technologies, Burlington, ON, Canada) three times and trypsinized for 3 min with 150 μL of trypsin-EDTA (disodium ethylenediaminetetraacetic acid) 0.25% (Cat#25200-056, Life Technologies, Burlington, ON, Canada). A total of 500 μL of DMEM/F12 without phenol red was added to harvest cells. From each sample, 10 μL of cell suspension was mixed with 10 μL of 0.4% Trypan Blue dye (Cat#15250061, Life Technologies, Burlington, ON, Canada), incubated for 5 min, and then counted on a hemocytometer. At least three independent experiments were conducted with two technical replicates per condition. The results are represented as the % cell viability ([Live cells/Total cell count] × 100) and the percentage of relative survival (% relative survival = [Number of live cells/cm^2^ of samples]/[number of live cells/cm^2^ of the control] × 100). Data were analyzed using a Kruskal–Wallis test with a Dunnett’s post hoc to compare each exposure vs. the matched negative control in SigmaPlot 14.5 (Systat Software Inc., San Jose, CA, USA).

### 2.8. CometChip Assay

The Trevigen’s 96-well CometChip^®^ system (Cedarlane Laboratories, Burlington, ON, Canada) was used as a high-throughput platform to expose cells, electrophorese, and fluorescently measure DNA damage. The comet assay was performed according to a previous study [34].

In brief, the CometChip (Cat#4260-096-01, Cedarlane Laboratories, Burlington, ON, Canada) was equilibrated in 100 mL DPBS at room temperature for 30 min, following which 100 μL of single-cell suspension (150,000 cells/mL in DMEM/F12 without phenol red) was loaded into each well of the 96-well plate and the CometChip was incubated for 20 min. Post-incubation, media was removed, and 50 μL of DMEM/F12 without phenol red was added to each well. MONPs suspensions, MOMPs suspensions, and dissolved metal analog solutions in the cell culture medium were prepared at 2× concentrations, and 50 μL of suspension was added to individual wells to obtain a final concentration of 1× in a final volume of 100 μL. For the negative control, cells were incubated in the cell culture medium for 4 h, and for the positive control, cells were exposed for 1 h to 100 μM H_2_O_2_ (Cat#216763, Sigma–Aldrich, Oakville, ON, Canada). After 2 and 4 h of exposure, the medium was removed from wells, the CometChip was washed with DPBS and overlaid with 5 mL of LMAgarose (Cat#4250-500-02, Cedarlane Laboratories, Burlington, ON, Canada), which solidified after 3 min at room temperature and 12 min at 4 °C. The CometChip with embedded cells was incubated in 100 mL of lysis solution (Cat#4250-050-01, Cedarlane Laboratories, Burlington, ON, Canada) for 1 h at 4 °C and equilibrated for 20 min two times in 250 mL of alkaline solution (pH > 13, 200 mM sodium hydroxide [NaOH] [Cat#S8045-500, Sigma–Aldrich, Oakville, ON, Canada], 1 mM EDTA [Cat#15575-020, Life Technologies, Burlington, ON, Canada], and 0.1% Triton-X [Cat#BP-151-500, Fisher Scientific, Whitby, ON, Canada]). The electrophoresis was performed at 4 °C for 50 min at a constant 21 V (variable current 280 mA) using 700 mL of alkaline solution. The CometChip was neutralized 2 times for 15 min in 100 mL of 400 mM Tris-HCl buffer (prepared from 1 M pH 7.4 Tris-HCl [Cat#T2194-1L, Sigma–Aldrich, Oakville, ON, Canada]), and then in 100 mL of 20 mM Tris-HCl for 30 min (prepared from 1 M pH 7.4 Tris-HCl). The neutralized chip was stained for 14 h in 100 mL 0.2X SYBR^®^ Gold (Cat#S11494, Life Technologies, Burlington, ON, Canada) in 20 mM Tris-HCl. After that, the CometChip was destained for 1 h at room temperature in 100 mL 20 mM Tris pH 7.4 and kept in 100 mL of DPBS. TIFF Images were generated using a Leica DMI8 automated confocal fluorescence microscope (Leica Microsystems, Wetzlar, Germany) at 5× magnification and were analyzed using the Trevigen Comet Software 1.3d (Bio-Techne, Devens, MA, USA). Artifacts were excluded from each well (double comet, debris, etc.).

Wells with more than 50 comets were included in the final analysis, except for cells treated with high concentrations (50 µg/mL or 100 µg/mL) of CuO NPs, MnO_2_ NPs, NiO NPs, Fe_2_O_3_ NPs, MnO_2_ MPs, and Fe_2_O_3_ MPs, which resulted in non-uniform or incomplete staining of comet heads (Appendix A). This resulted in less than 50 comets/well. Thus, for high-concentration exposures, a decision was made to count the wells with less than 50 comets. Each experiment considered a minimum of 2 and a maximum of 8 technical replicates, with at least three independent experiments performed. The mean percentage of DNA in the tail was used as the metric for DNA damage. Data were normalized via a log transformation and passed the normality test (Shapiro–Wilk). The transformed datasets were analyzed using a one-way analysis of variance (ANOVA) with a Dunnett’s post hoc to compare each exposure to the respective negative control in SigmaPlot 14.5.

### 2.9. In Vitro MicroFlow^®^ MN Assay

A total of 5000 FE1 cells were seeded in each well of a 96-well plate using DMEM/F12 cell culture medium without phenol red and were incubated at 37 °C in a humidified atmosphere of 5% CO_2_. Following 24 h incubation, cells were exposed to varying concentrations of MONPs, MOMPs, or dissolved metal analogs for 40 h. According to the MicroFlow MN assay, cells should be exposed for a duration that approximates 1.5 to 2 normal cell cycles to evaluate the MN induction. The doubling time of FE1 cells is 17 h; therefore, 40 h of exposures reflected cell division over 2 normal cell cycles. For the negative control, cells were incubated for 40 h in the cell culture medium, whereas for the positive control, cells were exposed to 500 µM of methyl methanesulfonate (MMS) (Cat# 129925, Sigma–Aldrich, Oakville, ON, Canada) for 40 h. The MicroFlow kit (In Vitro MicroFlow kit, 1000/200, 96 well; Litron Laboratories, Rochester, NY, USA) was used to measure cytotoxicity, fold increase of apoptotic/necrotic cells, and MN frequency using flow cytometry. Sample preparation, staining, and flow cytometry analysis were performed according to the manufacturer’s instructions. In brief, following the 40 h exposure, cells were washed with DMEM/F12 cell culture medium without phenol red and placed on ice for 20 min. The supernatant was removed, and 50 µL of freshly prepared Complete Nucleic Acid Dye A was added to each well. The plate was placed under a visible light source on ice for 30 min, and 150 µL of cold 1X Buffer Solution was added to each well. The supernatant was carefully removed. Cells were lysed, stained with 100 µL of Complete Lysis Solution 1, incubated at 37 °C for 1 h, and then in 100 µL of freshly prepared Completed Lysis Solution 2 at room temperature for 30 min. A Becton–Dickinson LSRFortessa^TM^ 5 laser analyzer (Becton–Dickinson, San Jose, CA, USA) was used to collect data. An analysis stop gate of 5000 ethidium monoazide (EMA)-negative nuclei was applied. The % relative survival, the increase of apoptotic/necrotic cells (EMA fold increase), and the MN frequency (% MN) were determined [52].

In brief, % relative survival was determined using intact viable nuclei-to-bead ratios in exposed cells versus the negative control by spiking the cell suspensions with 6-micron fluorescent microspheres (Cat#C-16508, Life Technologies, Burlington, ON, Canada), which function as the internal standards. Apoptotic/necrotic cells were stained using EMA dye (contained in the kit), which crosses the outer membrane of necrotic and apoptotic cells. EMA fold increase was determined using % Parent apoptotic/necrotic in exposed cells versus the average of % Parent apoptotic/necrotic in the negative control. % MN was scored in exposed cells and negative control using the double staining procedure outlined in the instruction manual. % MN was calculated as % MN = (MN Events/Nucleated Events) × 100. Then, % MN was converted to MN-fold increase values relative to negative controls [53]. The final data represents 3–4 biological replicates per condition. Statistical significance was determined using the Student’s *t*-test, which was used when both groups passed the normality test (Shapiro–Wilk) and equal variance test (Brown-Forsythe). For groups that were normally distributed (Shapiro–Wilk) but did not pass the equal variance, a Welch’s *t*-test was conducted. Mann–Whitney Rank Sum Test was conducted for some groups that did not pass the normality and equal variance tests. The statistical analysis was performed in SigmaPlot 14.5.

### 2.10. Correlation Analysis

To evaluate whether significant monotonic relationships were present between the two genotoxicity endpoints (% DNA in the tail, MN-fold increase) and the physicochemical properties of the MO tested, a Spearman’s rank-order correlation analysis was conducted. For this purpose, the response at the highest concentration at the latest timepoint for each MO form was used for % DNA in the tail, whereas the response at the highest concentration for each MO form where % relative survival was 40% or higher was used for MN-fold increase. The data from both genotoxicity endpoints were normalized to the concentration of MO expressed in terms of µM of constituent metal. The responses were correlated with nine physicochemical properties: % solubility at 10 µg/mL, % solubility at 100 µg/mL, valence state of the constituent metal, atomic mass of the metal constituent (g/mol), primary particle size (nm), SSA (m^2^/g) experimentally determined by BET, and HD (nm), PDI, and ZP (mV) in cell culture medium. A correlation was considered significant if *p* < 0.05.

### 2.11. Benchmark Concentration Modeling (BMC)

BMC modeling of the % DNA in the tail and % MN responses was conducted using PROAST (RIVM National Institute for Public Health and the Environment). The Benchmark response (BMR), which is a predetermined change in response as compared to negative control, was set to the commonly used value of 10% relative risk. Data with BMC/lower 95% confidence interval of the BMC (BMCL) > 20, BMC with values higher than the highest concentration, and samples with a non-significant trend according to the Akaike information criterion were removed.

All materials previously evaluated and reported in Boyadzhiev et al. [34], uncoated ZnO NPs (US3580), uncoated CuO NPs (544868), uncoated TiO_2_ NPs, silica-coated TiO_2_ NPs (5422HT), silica and alumina-coated TiO_2_ NPs (5423HT), silica and stearic acid-coated TiO_2_ NPs (5424HT), silica and silicone oil coated TiO_2_ NPs (5425HT), uncoated TiO_2_ NPs (MKA050), uncoated rutile TiO_2_ NPs (MKNR050P), uncoated TiO_2_ NPs (MKNA005), TiO_2_ nanowires (774510), ZnO MPs, CuO MPs, TiO_2_ MPs, ZnCl_2_ and CuCl_2_ were included in the analysis.

## 3. Results

### 3.1. MONPs Characterization

The primary size determined by TEM, the HD, PDI, and ZP in the cell culture medium are summarized in Table 3. Part of the characterization of uncoated ZnO NPs (US3580), uncoated CuO NPs (544868), uncoated NiO NPs (US3355), uncoated Al_2_O_3_ NPs (544833), uncoated TiO_2_ NPs, ZnO MPs, CuO MPs, NiO MPs, Al_2_O_3_ MPs, and TiO_2_ MPs were previously reported in Boyadzhiev et al. [34,49,54] and are indicated with letters “a”, “b”, and “c” in Table 3. The characterization of HD, PDI, and ZP in dH_2_O is found in Appendix A, and the BET SSA characterization is found in Appendix A. The frequency distribution of the MONPs can be found in Appendix A. The Appendix A shows representative SEM images of MnO_2_ MOMPs, CeO_2_ MOMPs, and Fe_2_O_3_ MPs. The images of the previously reported MONPs and MOMPs can be found in Boyadzhiev et al. [34,54].

The lengths of all ZnO NPs ranged from 23.9 ± 7.2 to 53.51 ± 38.5 nm, whereas the lengths of CuO NPs ranged from 63.07 ± 34.05 to 84.21 ± 55.46 nm. The size of uncoated MnO_2_ NPs (4910DX) was from 36.06 ± 34.88 nm. All NiO NPs exhibited similar lengths from 25.56 ± 13.16 nm to 31.99 ± 14.7 nm. Uncoated Al_2_O_3_ NPs (544833) exhibited 23.92 nm ± 11.84 nm size (reported in Boyadzhiev et al. [54]). Uncoated CeO_2_ NPs (US3136) showed 39.78 ± 14.55 nm, and the rest of the CeO_2_ NPs (Uncoated-US3036, PVP-coated and stearic acid-coated [US3037]) exhibited smaller lengths, ranging from 10.61 ± 2.71 to 14.12 ± 4.54 nm. The uncoated TiO_2_ NPs were 26.8 ± 8.9 nm in length (reported previously [34,54]). Uncoated Fe_2_O_3_ NPs (US3160) were 30.95 ± 13.12 nm (Table 3). Almost all MONPs had an aspect ratio spanning from 1.22 ± 0.20 to 1.39 ± 0.39 and showed a spherical morphology, except uncoated MnO_2_ NPs (4910DX) and uncoated Al_2_O_3_ NPs (544833), which exhibited an aspect ratio of ~2.7 with a rod-like shape (reported previously in Boyadzhiev et al. [54] and Appendix A).

The characterization of MONPs in dH_2_O showed HD and PDI values ranging from ~134 to 647 nm and from ~0.14 to 0.5, respectively, suggesting that all of them were polydisperse suspensions (Appendix A). The HD and PDI in the cell culture medium were closer to the values observed in water, which ranged from ~151 to 494 nm and ~0.19 to 0.56, respectively (Table 3). The decrease in ZP values to ~−12 mV and their polydispersity state in the cell culture medium is expected due to the cell culture medium components and its high osmolarity (Table 3) [43].

The SSA as determined by BET was from 16.52 to 28.28 m^2^/g for ZnO NPs, 6.11–10.34 m^2^/g for CuO NPs (showed the lowest SSA), 42.16 m^2^/g for uncoated MnO_2_ NPs (4910DX), 14.63 to 36.68 m^2^/g for NiO NPs, 145.3 m^2^/g for uncoated Al_2_O_3_ NPs (544833), which was the highest of all the MONPs, 14.06 to 67.11 m^2^/g for CeO_2_ NPs, and 52.73 and 44.88 m^2^/g for uncoated TiO_2_ NPs and uncoated Fe_2_O_3_ NPs (US3160), respectively (Appendix A). From SSA analysis of the MOMPs, CuO MPs showed the lowest value of SSA with 0.797 m^2^/g, whereas Al_2_O_3_ MPs showed the highest value with 22.32 m^2^/mg. The rest of the MOMPs exhibited SSA values ranging from 2.52 to 10.8 m^2^/g (Appendix A). The SSA experimentally determined by BET for all TiO_2_ variants [34] included in the BMC analysis is found in Appendix A.

### 3.2. Cell Viability and Relative Survival

Cell viability and relative survival were evaluated by the Trypan Blue exclusion method as indicators of cytotoxicity following exposure to different concentrations of MONPs, MOMPs, and dissolved metal analogs for 2 and 4 h. Uncoated MnO_2_ NPs (4910DX), MnO_2_ MPs, MnSO_4_, uncoated Fe_2_O_3_ NPs (US3160) and Fe_2_O_3_ MPs were evaluated after 24 h of exposure because their NP and MP forms did not induce a decrease in cell viability, no cytotoxicity evaluation was conducted at 2 and 4 h. The % cell viability results of uncoated ZnO NPs (US3580), ZnO MPs, ZnCl_2_, uncoated CuO NPs (544868), CuO MPs, CuCl_2_, uncoated TiO_2_ NPs, and TiO_2_ MPs were reported previously in Boyadzhiev et al. [34] and are included here for comparison purposes. The term “variants” refers to uncoated or coated MONPs and MOMPs.

The highest concentration evaluated for ZnO variants in this study was set to 20 µg/mL, as higher concentrations than this induced cell rounding and significant loss in viability at 4 h post-exposure in a previous study [34]. Similar cytotoxicity was not observed after exposure to other materials tested in the current study. Percent cell viability did not alter after exposure to MO variants and dissolved metal analogs at any concentrations or post-exposure time points (Appendix A), except for uncoated Fe_2_O_3_ NPs (US3160), which showed 93% and 89% viability at 50 and 100 µg/mL after 24 h compared to their matched negative control (Appendix A).

With regards to % relative survival, although a decreasing trend was observed at the highest concentration of 20 µg/mL of the uncoated or coated ZnO NPs or 33 µg/mL of ZnCl_2_, none of them induced a statistically significant decrease compared to the matched negative control at 2 or 4 h (Appendix A), the exposure durations used for the comet assay. With respect to Cu, Mn, and Ni, a majority of variants and dissolved metal analogs tested did not induce a decrease in % relative survival at any of the timepoints except for MnSO_4_, which induced reduction in the relative survival to 60% after 24 h of exposure (Appendix A) and stearic acid-coated NiO NPs (US3352), NiO MPs and NiCl_2_ which exhibited a tendency to affect the % relative survival only at 4 h (Appendix A). Uncoated Al_2_O_3_ NPs (544833), Al_2_O_3_ MPs, and AlCl_3_ exposure did not impact the % relative survival after 4 h; therefore, no cytotoxicity evaluation was conducted at 2 h of exposure (Appendix A). However, exposure to stearic acid-coated CeO_2_ NPs (US3037) resulted in a significant decrease in % relative survival (71%) at 134 μg/mL after 4 h post-exposure (Appendix A). Since uncoated CeO_2_ NPs (US3036) did not decrease % relative survival after 24 h of exposure, this NP was not evaluated at an earlier post-exposure time point. CeCl_3_ did not change the % relative survival with respect to the matched negative control (Appendix A).

After 2 and 4 h of exposure to uncoated TiO_2_ NPs, the % relative survival did not decrease at any concentration (Appendix A). The exposure for 24 h to Fe_2_O_3_ variants did not induce any statistically significant decrease at any concentration after 24 h of exposure; thus, no evaluation was conducted at 2 or 4 h (Appendix A).

### 3.3. DNA Strand Break Induction

Only the statistically significant results are described based on differences between the exposed samples and the matched negative controls.

#### 3.3.1. ZnO Variants and ZnCl_2_

Only stearic acid-treated ZnO NPs (8412DL) induced an increase in % DNA in the tail (6.7%) at 10 μg/mL at 2 h. All other ZnO NP variants, except the ZnO MPs, induced ~8–15% DNA in the tail compared to the matched negative controls (~4–5%) at the higher concentration of 20 μg/mL. At 4 h, uncoated ZnO NPs (US3580), uncoated ZnO NPs (5811HT), APTES-coated ZnO NPs (5812HT), and stearic acid-treated ZnO NPs (8412 DL) showed 13, 11, 12 and 24% DNA in the tail, respectively, at 10 μg/mL, and, at 20 μg/mL, the % DNA in the tail increased to 36, 33, 46 and 33%, respectively (Figure 1). ZnO MPs and ZnCl_2_ exposure induced DNA strand breaks after 4 h of exposure only at 20 µg/mL and 33 µg/mL with 29 and 15% DNA in the tail, respectively (Figure 1).

#### 3.3.2. CuO Variants and CuCl_2_

Among all CuO NPs, uncoated CuO NPs (544868, US3070) exhibited significant DNA damage at the lowest dose of 5 μg/mL with ~6–6.8% and ~10–12% DNA in the tail at 2 and 4 h, respectively, compared to matched negative controls (~4.6%). Uncoated CuO NPs (544868) caused 6, 8, 14, and 24% DNA in the tail at 2 h and 10, 17, 34, and 58% DNA in the tail at 4 h at the concentrations of 5, 10, 25, and 50 μg/mL, respectively. A similar response was observed for uncoated CuO NPs (US3070) with 7, 10, 22, and 28% DNA in the tail at 2 h and 12, 25, 42, and 51% at 4 h after exposure to 5, 10, 25 and 50 μg/mL, respectively. PVP-coated CuO NPs (US3070) and silane-coated CuO NPs (US3070) showed 7, 14, and 19–24% at 2 h after exposure to 10, 25 and 50 µg/mL. At 4 h, both PVP and silane-coated CuO NPs induced 9, 15, ~ 30, and 38% DNA in the tail at 5, 10, 25, and 50 μg/mL concentrations, respectively (Figure 2). CuO MPs and CuCl_2_ did not induce DNA damage (Figure 2).

#### 3.3.3. MnO_2_ Variants

Uncoated MnO_2_ NPs (4910DX) induced 14 and 22% DNA in the tail after 2 h of exposure and 22 and 34% after 4 h of exposure to 50 and 100 µg/mL. MnO_2_ MPs also induced DNA breaks with up to 13% DNA in the tail at 100 µg/mL at 4 h, compared to the matched negative controls (~4.58%) (Figure 3). MnSO_4_ did not increase the % DNA in the tail at any concentrations tested (Appendix A). Concentrations higher than 15 µg/mL of MnSO_4_ were not evaluated at 4 h because 15 µg/mL showed high cytotoxicity at 24 h (60% relative survival [Appendix A]).

#### 3.3.4. NiO Variants and NiCl_2_

At 2 h post-exposure, DNA damage was evident in cells treated with uncoated/coated NiO NPs and after treatment with NiO MPs, showing 7–9% of DNA in the tail at 5 and 10 μg/mL and 16–27% at 25, 50 and 100 μg/mL. At 4 h post-exposure, these particles showed 9–17% of DNA in the tail at 5 and 10 μg/mL and 15–26% at 25, 50 and 100 μg/mL concentrations. The exception was PVP-coated NiO NPs (US3352), which did not induce significant DNA strand breaks at 5 or 10 μg/mL at both the time points tested. Similarly, NiCl_2_ did not induce a response compared to the matched negative controls at 2 or 4 h (~5%) (Figure 4).

#### 3.3.5. Al_2_O_3_ Variants and AlCl_3_

Regarding Al_2_O_3_ variants (uncoated Al_2_O_3_ NPs [544833], Al_2_O_3_ MPs) and AlCl_3_, all of them showed values from 4 to 7% of DNA in the tail, but there was no statistically significant difference versus the negative controls (~5% DNA in the tail) (Figure 5).

#### 3.3.6. CeO_2_ Variants and CeCl_3_

For CeO_2_ variants and CeCl_3_, the negative controls showed ~5% of DNA in the tail. A subtle but significant increase in % DNA in the tail was observed after exposure to uncoated CeO_2_ NPs (US3136) at 100 µg/mL after 2 h and 4 h (~6.7%) and CeO_2_ NPs (US3036) at the same concentration at 2 h (6.6%). The PVP-coated CeO_2_ NPs (US3037) increased % DNA in the tail to 6.8–9.5% at 50 and 100 µg/mL at 2 h and to 6–7.14% at 10, 25, 50, and 100 ug/mL at 4 h of exposure. Stearic acid-coated CeO_2_ NPs (US3037) showed ~5–5.7% DNA in the tail at 3, 7, 17, and 134 µg/mL at 2 h and 5–5.8% at 4 h following exposure to 34, 67, and 134 µg/mL concentrations. However, a concentration or time response for all uncoated and coated CeO_2_ NPs was not observed. CeO_2_ MPs and CeCl_3_ did not induce DNA breaks (Figure 6).

#### 3.3.7. Fe_2_O_3_ Variants

Uncoated Fe_2_O_3_ NPs (US3160) induced a slight significant increase after 2 and 4 h of exposure at 100 µg/mL of 5.9% DNA in the tail. Fe_2_O_3_ MPs did not increase the % DNA in the tail compared to the matched negative controls (4.4%) (Figure 7).

Appendix A show representative comet images from MO variants that induced higher levels of % DNA in the tail (ZnO, CuO, MnO_2,_ and NiO). The images from the previously published data can be found in Boyadzhiev et al. [34]. The results for all TiO_2_ variants were previously published in Boyadzhiev et al. [34]. In brief, a minimal response of ~10% DNA in the tail was observed after exposure to silica-coated TiO_2_ NPs (5422HT), silica and alumina-coated TiO_2_ NPs (5423HT), and uncoated rutile TiO_2_ (MKNR050P) NPs.

### 3.4. MN Induction

Cells were exposed to different concentrations of MO variants for 40 h, and % relative survival and MN-fold increase relative to the unexposed controls were determined (Figure 8). Some exposures induced high toxicity, reducing the relative survival to less than 40%, which could contribute to false positive results. These results are denoted by the letter “a” in the bar graphs.

#### 3.4.1. ZnO Variants and ZnCl_2_

Uncoated ZnO NPs (US3580) induced 2.3- and 2.7-MN-fold increases at 1 and 3 µg/mL, whereas APTES-coated ZnO NPs (5812HT) induced a 2.5-MN-fold increase only at 3 µg/mL and stearic acid-coated ZnO NPs (8412DL) induced 2.7-, 3-, and 3.9-MN-fold increases at 1, 2 and 3 µg/mL (Figure 8A). ZnO MPs did not induce MN formation (Figure 8A), whereas ZnCl_2_ induced a significant increase of 1.8-MN-fold increase only at 1.6 µg/mL (Figure 9A). The highest concentration of all ZnO variants and ZnCl_2_ showed the highest level of cytotoxicity with less than 40% relative survival, and thus, the MN induction results are not reported for this concentration (Figure 8A and Figure 9A).

#### 3.4.2. CuO Variants and CuCl_2_

With respect to CuO variants, uncoated CuO NPs (544868) induced the highest response at the lower concentrations of 2 and 4 µg/mL with 3- and 3.8-MN-fold increase, and at 8 µg/mL, a 4.6-MN-fold increase was observed. However, the 8 µg/mL concentration also induced high cytotoxicity (32% relative survival). PVP-coated CuO NPs (US3070) induced a response only at 8 µg/mL (1.6-MN-fold increase) concentration. Silane-coated CuO NPs (US3070) showed 2.1- and 2-MN-fold increases at 4 and 8 µg/mL, whereas CuO MPs caused 1.4, 2.4, and 2.5-MN-fold increases at 2, 4, and 8 µg/mL, respectively (Figure 8B). CuCl_2_ did not induce MN or cytotoxicity at any concentration (Figure 9B).

#### 3.4.3. MnO_2_ Variants

Uncoated MnO_2_ NPs (4910DX) showed 2, 2.6, 2.6 and 2.3-MN-fold increases at 2.5, 5, 10 and 20 µg/mL. MnO_2_ MPs induced MN only at the highest concentrations of 10 and 20 µg/mL with 1.8 and 2.3-MN-fold increases (Figure 8C). MnSO_4_ did not induce MN or cytotoxicity (Figure 9C).

#### 3.4.4. NiO Variants and NiCl_2_

Uncoated NiO NPs (US3355) showed 2.3- and 2.11-MN-fold increases at 5 and 10 µg/mL, while at 20 µg/mL, the induction of MN could not be reliably determined due to the high cytotoxicity (32% relative survival). PVP-coated NiO NPs (US3352) and stearic acid-coated NiO NPs (US3352) did not induce MN but decreased the % relative cell survival to ~30% at the highest concentrations of 20 µg/mL and 12 µg/mL, respectively (Figure 8D). NiO MPs induced 2.4- and 2.7-MN-fold increases at 10 and 20 µg/mL (Figure 8D). NiCl_2_ did not induce a response but showed high cytotoxicity at 30 and 60 µg/mL (34 and 16% relative survival, respectively) (Figure 9D).

#### 3.4.5. Al_2_O_3_ Variants and AlCl_3_

Uncoated Al_2_O_3_ NPs (544833) induced a 2.1-MN-fold increase at 20 and 40 µg/mL. Al_2_O_3_ MPs induced a 1.2 and 1.8-MN-fold increases at 20 and 40 µg/mL (Figure 8E), whereas AlCl_3_ caused a 2-MN-fold increase at 20 µg/mL (Figure 9E). A 3.2-MN-fold increase was observed after exposure to 200 µg/mL of AlCl_3_, but the cytotoxicity was greater than 60% (Figure 9E).

#### 3.4.6. CeO_2_ Variants and CelCl_3_

Uncoated CeO_2_ NPs (US3036), PVP-coated CeO_2_ NPs (US3037), stearic acid-coated CeO_2_ NPs (US3037), and CeO_2_ MPs did not induce MN formation (Figure 8F). A concentration-dependent increase (1.6- and 2.7-MN-fold increase) was observed after exposure to 1 and 40 µg/mL of CeCl_3_, respectively (Figure 9F). CeCl_3_ at 80 µg/mL induced no statistically significant increase (3.6-MN-fold increase) but with high cytotoxicity (32% relative survival) (Figure 9F).

#### 3.4.7. TiO_2_ Variants

Uncoated TiO_2_ NPs and TiO_2_ MPs did not cause MN formation at any concentrations tested, but the response decreased after exposure to TiO_2_ MPs in a concentration-dependent manner, indicating potential interference (Figure 8G). Although both TiO_2_ variants showed a decreasing trend in % relative survival, the cytotoxicity was not above 60% (Figure 8G).

#### 3.4.8. Fe_2_O_3_ Variants

Uncoated Fe_2_O_3_ NPs (US3160) showed a trend to decrease MN induction at all concentrations tested (potential interference) and induced a significant decrease relative to negative control only at 20 µg/mL, whereas Fe_2_O_3_ MPs induced a 1.8-MN-fold increase at the highest concentration of 40 µg/mL (Figure 8H).

The % relative survival (42) and the MN-fold increase (10.2) after exposure to 500 µM MMS (positive control) are shown in Figure 9G.

### 3.5. Spearman’s Correlation of Genotoxicity with Physicochemical Properties of MO

To assess whether monotonic relationships are present between the physicochemical properties of the MO assessed and genotoxicity, a Spearman correlation was conducted (Figure 10, Appendix A).

With respect to correlations between physicochemical properties, solubility in DMEM cell culture medium at 10 and 100 µg/mL was significantly positively correlated, as were the atomic mass of the constituent metal and HD and PDI in cell culture medium. Conversely, primary particle size was significantly negatively correlated with particle SSA (Appendix A).

When % DNA in the tail was expressed in terms of µM of constituent metal, it can be seen that response at the highest tested concentration at 4 h is significantly positively correlated with % solubility at 10 and 100 µg/mL (Figure 10), as well as ZP, and negatively correlated with the valence state of the constituent metal. Similarly, MN-fold increase at 40 h normalized to µM of constituent metal at the highest admissible concentration was significantly correlated with solubility at 10 and 100 µg/mL (but not ZP) and significantly negatively correlated with the valence state of the constituent metal. Neither genotoxicity endpoint correlated with particle size or SSA for the MO assessed; however, responses for both endpoints were strongly positively correlated with each other (Appendix A).

### 3.6. BMC Analysis and Potency Ranking of Metal Oxide Materials

BMC modeling was employed to determine differences in potency for samples showing a concentration-dependent response in both 4 h comet and 40 h MN endpoints. The responses were compared with concentration normalized to constituent metal (µg metal/mL, Figure 11) and SSA (cm^2^ particles/cm^2^ well plate, Figure 12).

When the response was normalized to the amount of constituent metal in the exposure medium, differences were seen between % DNA in the tail (Figure 11A) and % MN induction (Figure 11B). For both endpoints, there were no apparent trends related to specific surface coatings, and there was widespread overlap of BMC confidence intervals between different forms (NP, MP, ionic) as well as between different constituent metals (Zn, Cu, Ni, etc.). A trend was apparent for % DNA in the tail at 4 h, with compounds with the same constituent metal showing similar BMCs. Zn materials had BMCs in the range of 2.2–5.1 µg metal /mL; for Cu materials, the BMCs were in the range of 0.3–0.9 µg metal /mL; for Mn materials, the range was 2.3–6.4 µg metal /mL; for Ni materials, the range was 0.09–0.96 µg metal /mL; Al materials presented no concentration response; Ce materials presented a BMC range of 2.2–29.6 µg metal /mL and Ti particles had BMCs in the range of 2.7–17.1 µg metal /mL.

With respect to % MN induction, no trends were observed. Uncoated CuO NPs (544868) showed the lowest BMC of all tested materials (0.43 µg metal /mL), but its confidence intervals overlapped with numerous other samples. All CeO_2_, TiO_2_, and Fe_2_O_3_ particles tested did not induce a concentration-dependent response.

When concentration is expressed in terms of the surface area of the particles to the surface area of the cell culture well in which the exposure was conducted (Figure 12), confidence intervals largely overlapped between samples for both endpoints. With respect to % DNA in the tail at 4 h, a trend was apparent in the BMCs between NPs and MPs analogs. For ZnO, MnO_2_, NiO, and CeO_2_, uncoated MPs present lower BMCs (but with overlapping confidence intervals) than all respective NP variants when the response was expressed in particle surface area (cm^2^ particles/cm^2^ well plate). For CuO, Al_2_O_3_, and TiO_2_ MPs, a concentration response was not present, and therefore, they were not included in the analysis.

With respect to % MN induction, CuO, MnO_2_, and NiO MPs showed lower BMCs than NPs, with some overlap present between confidence intervals. For ZnO, MPs were in the range of NPs tested, and for Al_2_O_3_, CeO_2_, TiO_2_, and Fe_2_O_3_, this comparison was not possible as MPs did not present calculable BMCs.

Due to the fact that multiple sample types did not feature a prominent concentration response, which precluded the calculation of an accurate BMC, potency ranking based on the no observed effect concentration (NOEC) values was conducted for both endpoints, using the exposure concentration normalized to µg metal/mL medium (Appendix A) and cm^2^ particles/cm^2^ well plate (Appendix A).

When the concentration was normalized to the amount of constituent metal in the exposure medium, it was seen that for comet and MN endpoints, uncoated MONPs were as potent or more potent than uncoated MOMPs (Figure 11, Appendix A). It should be kept in mind, however, that both TiO_2_ and Fe_2_O_3_ particles showed interference with respect to MN scoring, as reductions in the response were observed with respect to exposed samples as compared to negative controls. It can also be observed that dissolved metal equivalents are among the least potent forms of the MO assessed with respect to DNA damage.

However, for MN formation, divalent dissolved metals (ZnCl_2_, CuCl_2_, NiCl_2_, but not MnSO_4_) were among the least potent forms assessed, whereas the trivalent dissolved metals (AlCl_3_, CeCl_3_) were the most potent forms assessed of the respective MO. Surface-coated NP variants typically presented reduced or equal potency compared to uncoated MONPs. Two notable exceptions include the stearic acid-coated NiO NPs (US3352) and stearic acid-coated CeO_2_ NPs (US3037), which showed higher potency to induce DNA and chromosomal damage compared to uncoated NPs, respectively. This difference was due to difficulties in suspending stearic acid-coated particles, reducing the expected exposure concentration delivered to cells and offsetting the NOEC relative to other variants. No other trends were apparent.

When the concentration was normalized to the SSA of the particles and surface area of the cell culture well, the trends for both endpoints changed (Figure 12, Appendix A). MOMPs presented equal or greater potency than uncoated MONPs at inducing DNA or chromosomal damage (except for CuO MPs for the 4 h comet endpoint). For coated MONP variants, potency was reduced compared to uncoated MONPs for both endpoints, except in the case of Zn materials. For ZnO NPs, APTES-coated ZnO NPs (5812HT) were more potent than uncoated ZnO NPs at inducing comet response (NOEC: 0.26 vs. 0.34 and 0.43 cm^2^ particles/cm^2^ well plate). As well, for TiO_2_ materials previously assessed, all coated varieties showed less potency than uncoated rutile TiO_2_ NPs (MKNR050P) but more potent than the three other uncoated varieties tested (uncoated TiO_2_ NPs, MKNA050, MKNA005).

## 4. Discussion

In the present study, the genotoxic potential of MONPs of different sizes, solubility, and surface coatings was evaluated in FE1 lung epithelial cells at different post-exposure time points and concentrations using the high-throughput CometChip and MicroFlow MN assays. Eight types of commercially acquired MONPs of varying solubility were included in the study—ZnO, CuO, MnO_2_, NiO, Al_2_O_3_, CeO_2_, TiO_2_, and Fe_2_O_3_. For each MONP type, respective bulk particles, surface coating variants, and soluble forms were included, where possible. The many varied forms of each metal oxide assessed allowed for the analysis of relationships between size, coating, solubility, and genotoxic potential.

### 4.1. Impact of Particle Size, Solubility, and Surface Coating on MONP Genotoxicity

The results of material dissolution testing conducted for the 8 MONPs used in this study in both water and DMEM cell culture medium have been previously published (Appendix A, Avramescu et al. [38,41]). Based on their dissolution in DMEM at 100 µg/mL, the materials were binned as highly soluble (>70% dissolution), moderately soluble (10–70% dissolution; CuO, ZnO NPs), low solubility (1–10% dissolution; NiO, MnO_2_), and negligibly soluble (<1% dissolution; CeO_2_, Al_2_O_3_, TiO_2_, Fe_2_O_3_). For some materials, dissolution in DMEM changed at lower concentration of 10 µg/mL, which showed that ZnO is highly soluble (94.5% dissolved), CuO is moderately soluble (12.6% dissolved), MnO_2_, Al_2_O_3_, and CeO_2_ are low soluble (4.79%, 1.11%, 1.12% dissolved, respectively), and NiO and TiO_2_ to be negligibly soluble particles (0.94%, 0.17%). A lower concentration was not assessed for Fe_2_O_3_ NPs (Appendix A).

Correlation analyses of 9 individual physicochemical properties with the 2 genotoxicity endpoints showed both endpoint responses are positively correlated with solubility in the cell culture medium at 10 and 100 µg/mL, indicating solubility is an important property for MO-induced genotoxicity (Figure 10). In addition to extracellular solubility, material solubility in acidic environments, such as in lysosomes, is also of key importance for MONPs due to their potential to localize to such compartments. Dissolution testing in Avramescu et al. [38] examined the solubility of ZnO, CeO_2_, and MnO_2_ NPs (also used in the present study) in artificial lysosomal fluid and noted that these materials exhibit high solubility, negligible solubility, and low solubility, respectively. For the other 5 MONPs examined, previous research using similar materials has indicated that CuO NPs are highly soluble [55], NiO NPs are moderate to highly soluble depending on the specific ENM characteristics [56], Al_2_O_3_ NPs are moderately soluble [57], and both TiO_2_ and Fe_2_O_3_ NPs are negligibly soluble in lysosomes [58]. All these studies indicate that the solubility pattern of the MONPs tested in this study is the same in cell culture medium and artificial lysosome fluid. While the solubility of specific MONP variants can change based on physicochemical properties, the results of the genotoxicity testing carried out in this study indicate that, in general, MONPs of negligible solubility in DMEM do not induce DNA strand breaks in the short term (Figure 1, Figure 2, Figure 3, Figure 4, Figure 5, Figure 6 and Figure 7), whereas MONPs of negligible solubility in lysosomal fluid do not lead to increases in MN induction (Figure 8). There was also a difference in the behavior of dissolved trivalent metals (Ce, Al) as compared to divalent metals (Zn, Cu, Ni) with respect to MN induction, where dissolved trivalent metals were more potent than NPs and MPs at inducing response, the dissolved divalent metals were less potent or showed similar potency to NPs and MPs analogs (Appendix A). Dissolution studies of Al_2_O_3_ and AlCl_3_ have shown that dissolved Al has the propensity to precipitate in neutral biological environments [38,59]. Similarly, while CeO_2_ is insoluble in lysosomes, dissolved cerium in the form of CeCl_3_ has been shown to localize to lysosomes where it forms an insoluble precipitate containing phosphorus [60]. It is possible that depletion of media resources due to precipitation of the dissolved fractions may explain MN induction in response to Al_2_O_3_ NPs, AlCl_3_, and CeCl_3_ and not by bivalent types. Together, these findings imply that in addition to MONP solubility, the stability of the dissolved form is another defining property governing their genotoxic potential, with instability in neutral and acidic environments leading to increases in DNA damage.

In addition to the impacts of solubility on MONP genotoxicity, this study also evaluated the impacts of particle surface coating on response. The particle surface mediates the first interaction between particles and cells or the biological milieu, defining the mode of internalization, the extent of distribution, accumulation, and eventual fate of NPs [61]. The surface of MONPs can be modified to protect the particles, decrease their state of agglomeration, increase particle stability, improve wettability, and increase their bioavailability [61]. Two coated variants of ZnO NPs (APTES, stearic acid-coated), two coated variants of CuO NPs (PVP, silane-coated), two coated variants of CeO_2_ NPs (PVP, stearic acid-coated), and three coated variants of NiO NPs (PVP, stearic acid, silane-coated) were examined for their potential to induce genotoxicity in lung epithelial cells. No separate solubility experiments were conducted for coated MONP variants. Based on both DNA strand break and MN induction, no pattern was observed with respect to the influence of specific surface coatings on toxicity (Figure 11 and Figure 12). Both PVP and silane surface coatings were shown to reduce DNA strand break and MN induced by exposure to uncoated CuO NPs (Figure 2 and Figure 8). A similar but more subtle response was observed with respect to NiO NPs, where stearic acid and PVP coatings reduced MN induction (Figure 8). In an in vivo study examining the pulmonary toxicity of paint-embedded TiO_2_ NPs, hyperspectral imaging showed they remain embedded in the paint matrix up to 28 days post-exposure, which was suggested as the reason for a decrease in their ability to alter gene expression and associated biological pathways as compared to non-embedded particles [62]. Surface coating of MONPs can protect their cores from dissolution, reducing ion leaching and toxicity [63,64]. This depends on the stability of the coating and the solubility of the MONPs in different microenvironments. Since no solubility testing was conducted with coated variants, it is possible that the observed reduction in genotoxicity after exposure to coated CuO and NiO NPs as compared to uncoated NPs could also be attributed to changes to particle surface-cell interaction and their solubility.

The study also examined the differences in genotoxic potential of MONPs and MOMPs. In general, NPs either induce the same or higher levels of both DNA strand break and MN induction as compared to matched MPs equivalents, with no significant correlation between primary particle size and genotoxicity noted (Appendix A). The only exception to this trend was with respect to NiO MPs, which showed a heightened DNA strand break response at 2 and 4 h as compared to NPs at a concentration of 100 µg/mL (Figure 4). However, this result was not observed at the lower concentrations. The observations from this study are in alignment with our previously published results [34] that showed DNA damage induced by ZnO, CuO, and TiO_2_ NPs with similar primary particle sizes was either higher or equal to their MPs analogs. Higher levels of genotoxicity with NPs compared to MPs are most commonly thought to result from their small volume and large SSA that increases the surface molecules available for interaction and, as a result, increases their surface reactivity, allowing for enhanced interaction of NPs with the biological medium, cellular membranes, and subcellular structures [10]. For materials such as CuO, particle size greatly influences the potential to induce DNA strand breaks, with MPs showing no response in the CometChip assay while both CuO NPs (544868 and US3070) induced pronounced concentration and time-dependent genotoxicity (Figure 2). A similar trend is seen with MnO_2_, with NPs leading to DNA strand break and MN induction at lower concentrations than what was observed for MPs. In addition, Al_2_O_3_ NPs showed significant MN induction at a concentration of 20 µg/mL, a response that was only observed at 20 and 40 µg/mL MPs concentration (Figure 8). For ZnO, CeO_2_, NiO, and TiO_2_, DNA strand break and MN induction were similar between MPs and NPs, with response mainly related to the specific metal in question (ex. Ce vs. Ni). These results suggested that while some MPs were not genotoxic, others showed delayed responses that also differed in magnitude. In recent years, some studies have shown that, in general, smaller NPs are more genotoxic than larger NPs or MPs [34,65,66]. For instance, in human umbilical vein endothelial cells (HUVECs) exposed to 1, 5, and 25 µg/mL SiO_2_ NPs of different sizes (10, 25, 50, and 100 nm) for 4 and 24 h, the observed DNA strand breaks at 4 h and MN induction at 24 h showed an inverse relationship with size, with higher responses recorded for smaller sized NPs [65]. In another study, polystyrene NPs induced more MN than polystyrene MPs at 50 and 100 µg/mL following 24 h exposure in A549 cells [66]. The results of the present study agree with the above observations, showing that the smaller primary particle size of MONPs vs. MOMPs contributes to an increased potential for DNA damage; however, chemical composition is also a critical property affecting genotoxicity.

### 4.2. Relative Potency Ranking of MONP Genotoxic Potential

As compared to classical points of departure (PODs), such as the lowest observed effect concentration (LOEC), NOEC, or lethal concentration 50 (LC_50_), BMC modeling can utilize the whole concentration-response curve to estimate potency. This can make it more accurate than classical PODs, but it requires a complete concentration response in order to have a precise estimation of the BMC and resulting confidence intervals. This means that for materials that do not show concentration-dependent responses, it is not possible to compute a BMC. In this study, BMC modeling was attempted in an effort to rank all compounds based on their genotoxic potency for both endpoints. Due to the large degree of overlap in BMC confidence intervals and the absence of calculable BMCs for numerous compounds, the ranking was also conducted using the NOEC (Appendix A).

When concentration was normalized to the amount of constituent metal in the exposure medium, MOMPs were less potent or as potent as MONPs at inducing DNA strand breaks and MN formation; however, this trend flipped when concentration was normalized to the SSA of the particles (Figure 11, Appendix A). The same trend was reported in a recent transcriptomic study examining 5 of the same uncoated MONPs (ZnO, CuO, NiO, Al_2_O_3_, TiO_2_), with potency trends based on SSA normalized concentrations showing that MPs are either as potent or more potent than corresponding MONPs [54]. In general, SSA is inversely proportional to particle size, with smaller particles showing larger surface areas. In this study, the primary particle size of NPs and MPs is significantly negatively correlated with SSA; however, both genotoxicity endpoints did not show a correlation with size or SSA (Appendix A), suggesting these properties may not be the defining ones for the toxicity response. A retrospective analysis of 9 individual in vivo studies involving rats and mice exposed to different types of NPs (polystyrene, TiO_2_, carbonaceous materials, Co_3_O_4_, Ni, ZnO, and nanoquartz) reported very strong correlations between the SSA dose and acute pulmonary toxicity [67]. However, a more recent analysis examining 88 individual in vivo studies including 13 different ENMs (NMs of Ag, carbon, Ce, cobalt, Cu, Fe, Ni, silicium, TiO_2_, and Zn) did not find such strong associations between SSA dose and resulting toxicity [68]. The results of the current study and the recent large-scale transcriptomic analysis, in addition to available in vivo data, imply that higher SSA may be associated with increased toxicity; however, MONP stability and other physicochemical properties discussed above can override this response.

### 4.3. Mechanisms Affecting Genotoxic Potential of MONPs

Based on the activity for each of the two genotoxicity endpoints (Figure 13), inferences can be derived regarding the type of DNA damage seen and potential mechanisms at play across the MONPs tested.

Based on the results of previously evaluated DNA damage response of ZnO, CuO and TiO_2_ NP, the dissolved component was determined to be the main mediator of DNA damage for ZnO NPs, whereas for CuO NPs, a “Trojan Horse” mechanism was considered more likely, and for TiO_2_ NPs surface interactions and subsequent ROS generation was suggested as a putative underlying mechanism [34]. The summary in Figure 13 shows that for Zn and Cu forms, their clastogenic/aneugenic activity is much the same across all sizes, coatings, and forms, except for ZnO MPs and CuCl_2_. In the case of ZnO MPs, a non-significant concentration-dependent increase was seen (Figure 8). Conversely, uncoated TiO_2_ NPs and MPs did not induce increases in MN induction, although interference of the assay was seen for TiO_2_ MPs and Fe_2_O_3_ NPs (Figure 8). For both CuO and ZnO NPs, genotoxicity is surmised to be due to ion homeostatic disturbances, with the dissolved fraction being the main mediator of ZnO NP response, whereas the nanoparticulate nature of CuO NPs is critical for its response. The positive MN call and negative comet call for CuO MPs indicate that larger-sized CuO particles require longer interaction time to induce DNA damage, potentially related to reduced uptake or reduced intracellular dissolution. For dissolved Cu, reduced uptake of the ionic form as compared to NPs and MPs is a possible explanation for its lack of response. Dissolved Cu is taken up via metal transporters, mainly through the high-affinity copper transporter known as hCTR1 in humans or Ctr1 in mice [69]. While not much is known about copper flux through murine Ctr1, hCTR1 is considered a relatively slow transporter with the ability to transport 5–10 Cu ions per second [69,70], compared to typical transporters which have transport rates of 100–10,000 molecules per second [71]. The slower rate of uptake of dissolved Cu as compared to CuO NPs could allow cells the time to respond to the stressor, reducing downstream toxicity resulting in a negative genotoxicity call (as described in Boyadzhiev et al. [34]).

For NiO, Al_2_O_3_, MnO_2_, CeO_2_, and Fe_2_O_3_ NPs not previously assessed, NiO and MnO_2_ NPs and MPs were DNA strand break and clastogen/aneugen positive, while the respective dissolved analog was negative for both. In mammalian cells, Mn functions as an essential micronutrient like Cu and Zn. In cellular model systems, elevated levels of Mn are associated with oxidative stress, mitochondrial dysfunction, and cell death [72]. From dissolution testing in Avramescu et al. [38], it can be seen that MnO_2_ NPs are soluble in both the cell culture medium as well as in artificial lysosomal fluid. It is surmised that genotoxicity resulting from MnO_2_ NP exposure proceeds according to the same mechanisms as CuO NPs, through a ‘Trojan Horse’ mechanism whereby the particles are internalized into the acidic compartments, followed by intracellular dissolution, resulting in disturbances of Mn homeostasis, oxidative stress, and subsequent DNA damage. The lower response of the MPs as compared to NPs in both comet and MN endpoints (Figure 3 and Figure 8) provides credence to this hypothesis, as MPs show lower dissolution than NPs at the same concentration in the cell culture medium (Avramescu et al. [38]). For dissolved MnSO_4_, the concentrations were chosen to represent extracellular dissolution, and from the lack of response seen, the extracellular dissolved component does not contribute to the genotoxicity induced by MnO_2_ NPs.

With respect to NiO NPs, the potential mechanisms are different. Ni is considered an essential micronutrient in certain prokaryotic organisms but not in mammals, and as such, no homeostatic system exists to specifically regulate its bioavailability. While NiO NPs are poorly to negligibly soluble in cell culture medium, they show solubility in lysosomal fluid [56]. Research on the genotoxicity of Ni compounds has shown that they function as weak mutagens via indirect mechanisms, with Ni ions showing preferential interaction with proteins as opposed to DNA [73]. The mechanism behind this toxicity involves the generation of ROS, inhibition of DNA repair enzymes, and epigenetic alterations, with carcinogenicity dependent upon the delivery of dissolved Ni to the nucleus [73]. From the summary in Figure 13, it can be seen that all Ni compounds, except for NiCl_2_, induce DNA strand breaks. Furthermore, both NiO NPs and MPs are positive for clastogenicity/aneugenicity, with non-significant upward trends seen for coated NPs and dissolved Ni. The lack of DNA strand break induction for NiCl_2_ in this study may be related to insufficient exposure time to allow for cellular uptake. From the comet responses, it can be surmised that NiO NPs and MPs are rapidly internalized into the cell, wherein dissolution or particle interaction begins to occur, resulting in DNA strand breaks as early as 2 h post-exposure (Figure 4). Due to the propensity for Ni to interact with proteins as well as induce DNA strand breaks, the positive MN call for uncoated NiO NPs could be due to either clastogenicity (accumulation of strand breaks culminating in chromosomal breakage) or aneugenicity (through interference with mitotic machinery).

In contrast, based on the negative DNA strand break and positive clastogenicity/aneugenicity calls, Al_2_O_3_ NPs seem to act indirectly with indications for a potential aneugenic mechanism. From the high-content transcriptomic analysis of the same Al_2_O_3_ NPs, MPs, and AlCl_3_ used in this study, minimal indications of oxidative stress and no indications of DNA damage responses were seen for up to 48 h of exposure [54]. Furthermore, it can be seen that Al_2_O_3_ NPs show low-negligible solubility in cell culture medium but can dissolve under acidic conditions in lysosomes [38,57]; however, dissolved Al is known to precipitate in the presence of biological material [38,59]. Evidence exists for the ability of Al to induce chromosomal aberrations in plants and to induce cross-linking of chromosomal proteins in mammals, but with little to no potential to induce DNA strand breaks, mutagenesis, or carcinogenicity [74]. Based on this information, it is proposed that the positive MN call and negative comet call for Al_2_O_3_ NPs are indicative of interference with mitotic machinery rather than accumulation of single- and double-stranded DNA breaks. This may be the result of the dissolved fraction, as AlCl_3_ was able to induce the formation of MN at much lower concentrations than Al_2_O_3_ NPs (Figure 11) or through direct interaction of NPs with mitotic machinery.

Finally, with respect to CeO_2_ NPs and Fe_2_O_3_ NPs, based on the results of both genotoxicity endpoints, these MONPs are considered to be non-genotoxic (Figure 13). For CeO_2_ NPs, minimal comet induction (<10%) and negative responses were seen in the MN assays; however, the dissolved CeCl_3_ was MN-positive. The lack of genotoxicity in CeO_2_ NPs is ascribed to limited surface reactivity as well as negligible solubility in both cell culture medium and lysosomal fluid [38], while dissolved Ce^3+^ may behave similarly to Al^3+^ as both are known to form phosphate-containing complexes in cells and cell culture medium. As seen for CeO_2_ NPs, TiO_2_ NPs, and Fe_2_O_3_ NPs are negligibly soluble in the cell culture medium as well as lysosomal fluid [38,58]. DNA strand break induction for both NPs and MPs is minimal (Figure 7); however, interference was seen for TiO_2_ NPs and Fe_2_O_3_ NPs in the MN assay, with a reduction in response across the concentration range (Figure 8). Thus, it was not possible to deduce the potential mechanistic differences conclusively, but the negative comet and MN call for the NPs suggest that they have a low surface reactivity, which could mitigate downstream toxicity resulting from particle interactions. In vivo studies from our lab have shown that TiO_2_ NPs induce inflammation in mouse lungs [24,75]. At low doses, TiO_2_ NPs have been shown to perturb pathways associated with ion homeostasis, which could be due to their tissue persistence and interference with the cellular cytoskeleton functions [75].

From the available literature, both negative and positive genotoxicity responses have been seen with regard to different types of Fe_2_O_3_ NPs and cell lines, with DNA strand break induction and oxidative DNA damage being the main mechanisms cited [76]. The Fe_2_O_3_ NPs assessed in this study do not appear to be reactive, based on comet results.

## 5. Conclusions

In conclusion, this study utilized the high-throughput CometChip and MicroFlow assays to assess the in vitro genotoxic potential of ZnO, CuO, MnO_2_, NiO, Al_2_O_3_, CeO_2_, TiO_2_, and Fe_2_O_3_, NPs, MPs, and dissolved metal analogs, with the purpose of identifying the role of solubility, particle size, and surface coating on response. Based on correlation analyses between concentration normalized endpoint responses and physicochemical properties, strong positive correlations were seen between particle solubility in cell culture medium and % DNA in the tail and MN induction across all tested particles, indicating that as solubility in medium increases, the propensity to induce genotoxicity increases as well. When concentration is normalized to constituent metal, MONPs exhibited the same or greater potency than MPs to induce both DNA strand breaks and MN formation; however, this trend reversed when concentration was normalized to SSA, indicating the larger surface area of NPs may at least partially explain the difference in potency between MONPs and MOMPs. Surface coating of MONPs inconsistently affected genotoxicity, with CuO and NiO NP coatings seen to reduce comet and MN responses. Differences in DNA strand break activity and clastogenicity/aneugenicity across the tested materials denote mechanistic differences in MONP response, which is related to both particle stability and chemical composition. The results of this study highlight that combinations of properties influence response to MONPs and that solubility alone, while playing an important role, cannot be solely accountable for observed toxicity. This may have implications on the potential application of read-across strategy in support of human health risk assessment of MO in their nano form.

## Figures and Tables

**Figure 1 nanomaterials-14-00743-f001:**
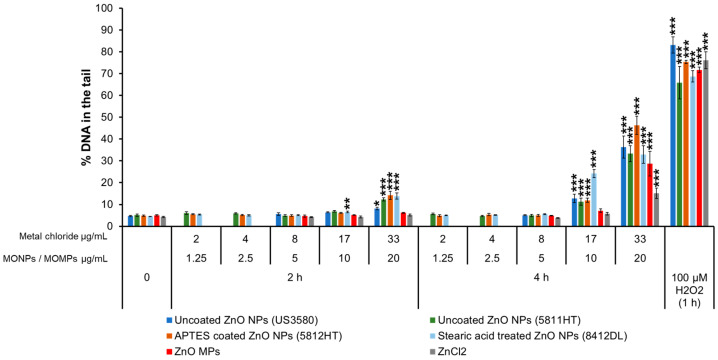
Percentage of DNA in the tail in FE1 cells after exposure to ZnO variants and ZnCl_2_ at 2 and 4 h. Data are presented as mean and standard error (*n* = 3–4). Statistically significant differences between the exposed samples and the matched negative control (4 h) were determined through one-way ANOVA with a Dunnett’s post hoc. * *p* < 0.05, ** *p* < 0.01, *** *p* < 0.001. Uncoated ZnO (US3580), ZnO MPs, and ZnCl_2_ data were previously reported [34].

**Figure 2 nanomaterials-14-00743-f002:**
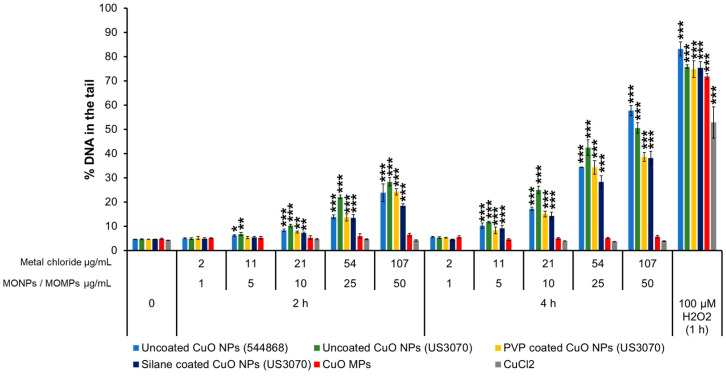
Percentage of DNA in the tail in FE1 cells after exposure to CuO variants and CuCl_2_ at 2 and 4 h. Data are presented as mean and standard error (*n* = 3–4). Statistically significant differences between the exposed samples and the matched negative control (4 h) were determined through one-way ANOVA with a Dunnett’s post hoc. * *p* < 0.05, ** *p* < 0.01, *** *p* < 0.001. Uncoated CuO NPs (544868), CuO MPs, and CuCl_2_ data were previously reported [34].

**Figure 3 nanomaterials-14-00743-f003:**
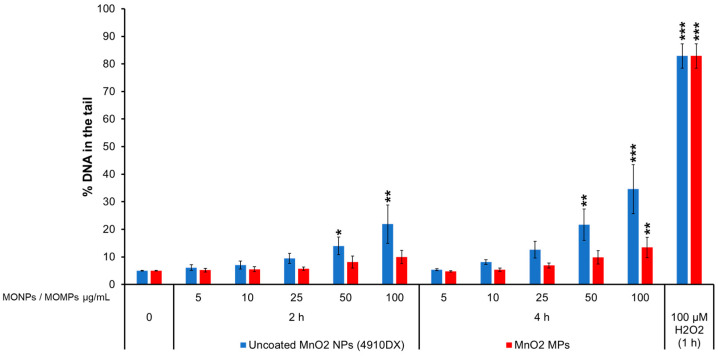
Percentage of DNA in the tail in FE1 cells after exposure to MnO_2_ variants at 2 and 4 h. Data are presented as mean and standard error (*n* = 3–4). Statistically significant differences between the exposed samples and the matched negative control (4 h) were determined through one-way ANOVA with a Dunnett’s post hoc. * *p* < 0.05, ** *p* < 0.01, *** *p* < 0.001.

**Figure 4 nanomaterials-14-00743-f004:**
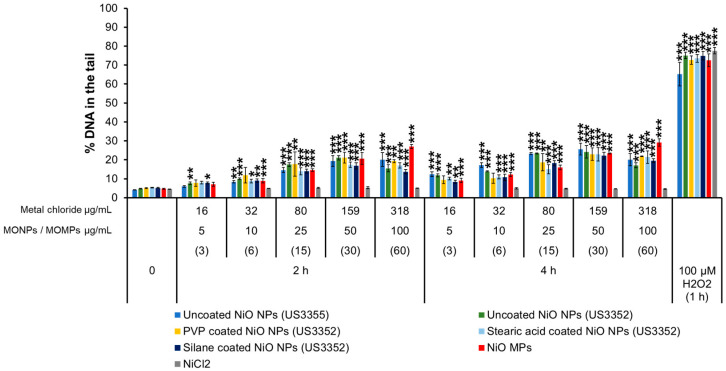
Percentage of DNA in the tail in FE1 cells after exposure to NiO variants and NiCl_2_ at 2 and 4 h. Data are presented as mean and standard error (*n* = 3–4). Statistically significant differences between the exposed samples and the matched negative control (4 h) were determined through one-way ANOVA with a Dunnett’s post hoc. * *p* < 0.05, ** *p* < 0.01, *** *p* < 0.001. Data in parenthesis indicates the concentration of stearic acid-coated NiO NPs.

**Figure 5 nanomaterials-14-00743-f005:**
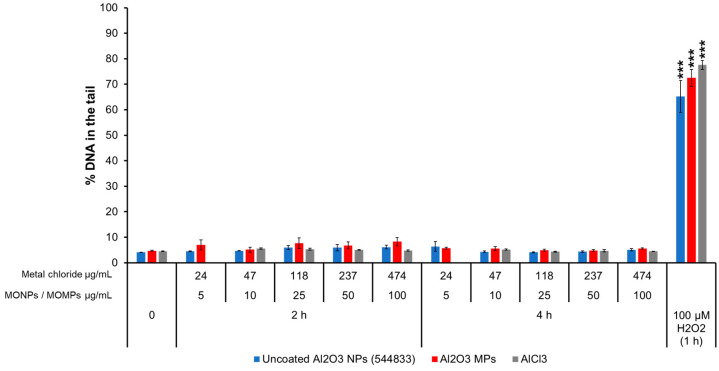
Percentage of DNA in the tail in FE1 cells after exposure to Al_2_O_3_ variants and AlCl_3_ at 2 and 4 h. Data are presented as mean and standard error (*n* = 3–4). Statistically significant differences between the exposed samples and the matched negative control (4 h) were determined through one-way ANOVA with a Dunnett’s post hoc. *** *p* < 0.001.

**Figure 6 nanomaterials-14-00743-f006:**
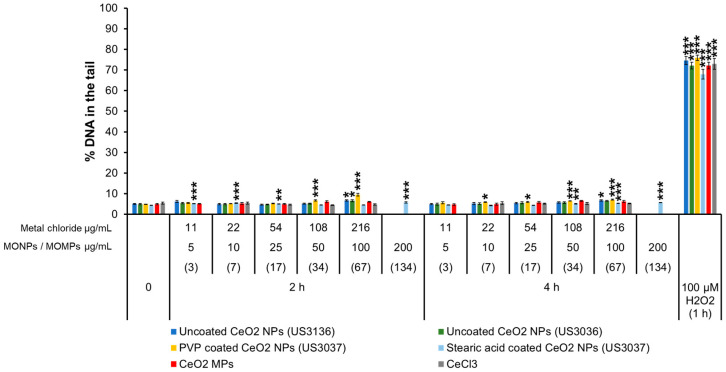
Percentage of DNA in the tail in FE1 cells after exposure to CeO_2_ variants and CeCl_3_ at 2 and 4 h. Data are presented as mean and standard error (*n* = 3–4). Statistically significant differences between the exposure and the matched negative control (4 h) were determined through one-way ANOVA with Dunnett’s post hoc. * *p* < 0.05, ** *p* < 0.01, *** *p* < 0.001. Data in parenthesis indicates the concentration of stearic acid-coated CeO_2_ NPs.

**Figure 7 nanomaterials-14-00743-f007:**
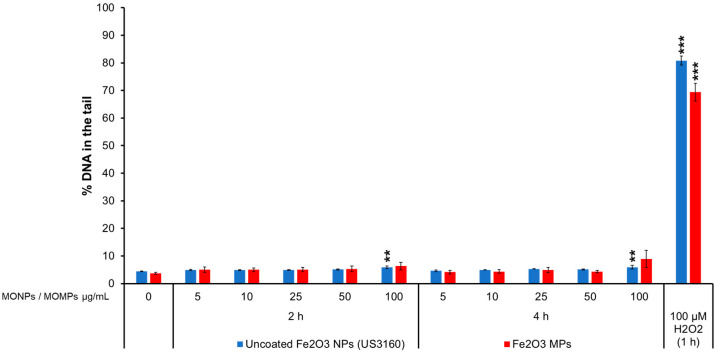
Percentage of DNA in the tail in FE1 cells after exposure to Fe_2_O_3_ variants at 2 and 4 h. Data are presented as mean and standard error (*n* = 3–4). Statistically significant differences between the exposed samples and the matched negative control (4 h) were determined through one-way ANOVA with a Dunnett’s post hoc. ** *p* < 0.01, *** *p* < 0.001.

**Figure 8 nanomaterials-14-00743-f008:**
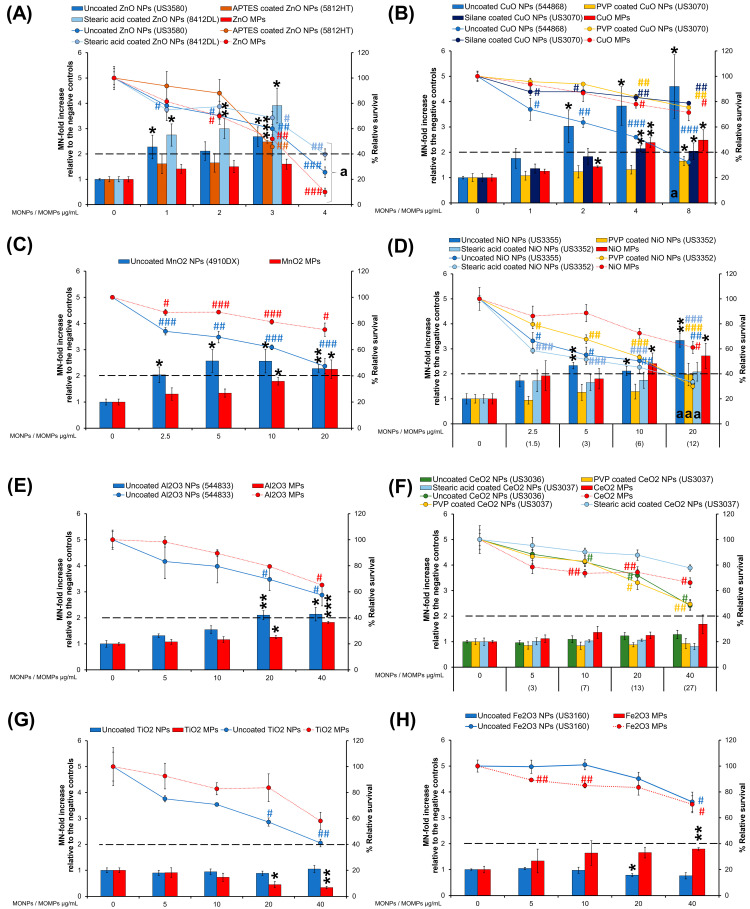
Cytotoxicity assessment (% relative survival) and MN-fold increase in FE1 cells after 40 h of exposure to MO variants. (**A**) ZnO variants, (**B**) CuO variants, (**C**) MnO_2_ variants, (**D**) NiO variants, (**E**) Al_2_O_3_ variants, (**F**) CeO_2_ variants, (**G**) TiO_2_ variants, (**H**) Fe_2_O_3_ variants. Data are presented as mean and standard error (*n* = 3–4). Data in parenthesis indicates the concentration of stearic acid-coated NiO NPs or stearic acid-coated CeO_2_ NPs. Statistically significant differences between the exposed samples and the matched negative controls were determined (see Section 2.9). % relative survival: ^#^
*p* < 0.05, ^##^
*p* < 0.01, ^###^
*p* < 0.001. MN-fold increase: * *p* < 0.05, ** *p* < 0.01, *** *p* < 0.001. The letter “a” indicates that the exposure was overtly cytotoxic, with relative survival dipping below 40%. The dashed line represents 40% of the relative survival threshold.

**Figure 9 nanomaterials-14-00743-f009:**
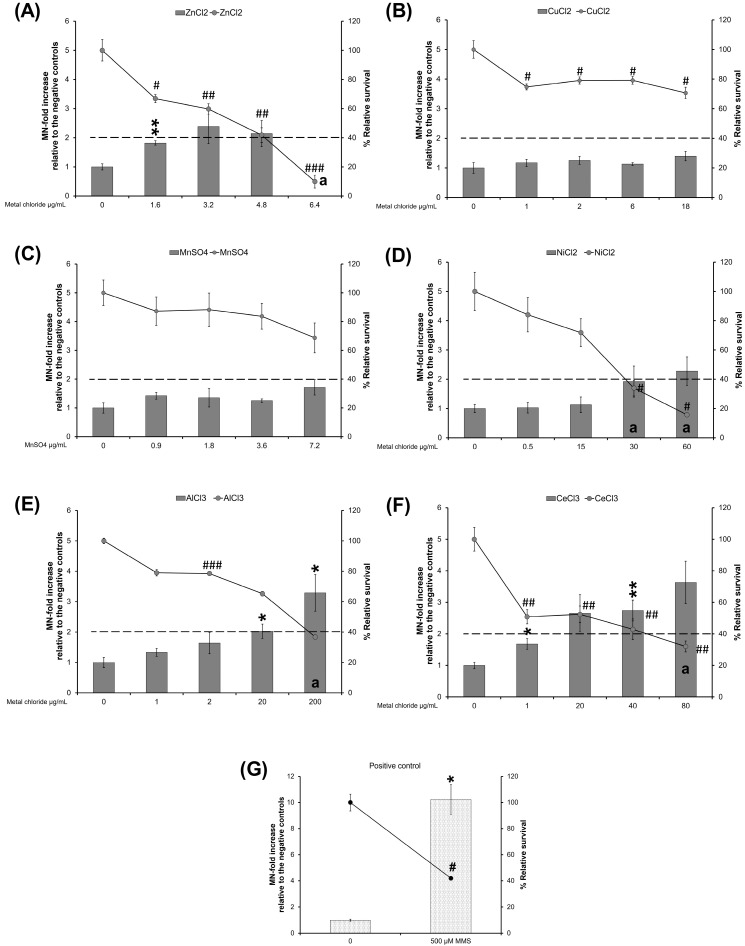
Cytotoxicity assessment (% relative survival) and MN-fold increase in FE1 cells after 40 h of exposure to dissolved metal analogs. (**A**) ZnCl_2_, (**B**) CuCl_2_, (**C**) MnSO_4_, (**D**) NiCl_2_, (**E**) AlCl_3_, (**F**) CeCl_3_, (**G**) Positive control: 500 µM MMS for 40 h. Data are presented as mean and standard error (*n* = 3–4). Statistically significant differences between the exposed samples and the matched negative controls were determined (see Section 2.9). % relative survival: ^#^
*p* < 0.05, ^##^
*p* < 0.01, ^###^
*p* < 0.001. MN-fold increase: * *p* < 0.05, ** *p* < 0.01. The letter “a” indicates that the exposure was overtly cytotoxic, with less than 40% relative survival. The dashed line represents 40% of the relative survival threshold.

**Figure 10 nanomaterials-14-00743-f010:**
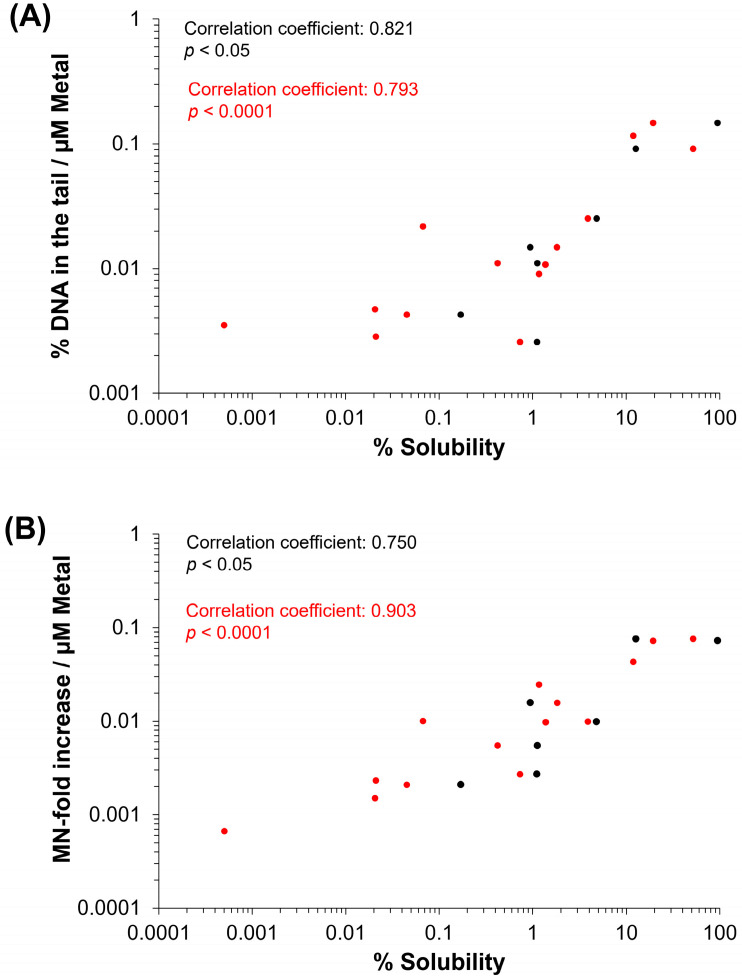
The relationship between solubility at 10 and 100 µg/mL, and (**A**) % DNA in the tail at 4 h and (**B**) MN-fold increase at 40 h when concentration is normalized to µM of constituent metal. The highest admissible concentration for each endpoint was used. Black circles: Solubility at 10 µg/mL (*n* = 7). Red circles: Solubility at 100 µg/mL (*n* = 14).

**Figure 11 nanomaterials-14-00743-f011:**
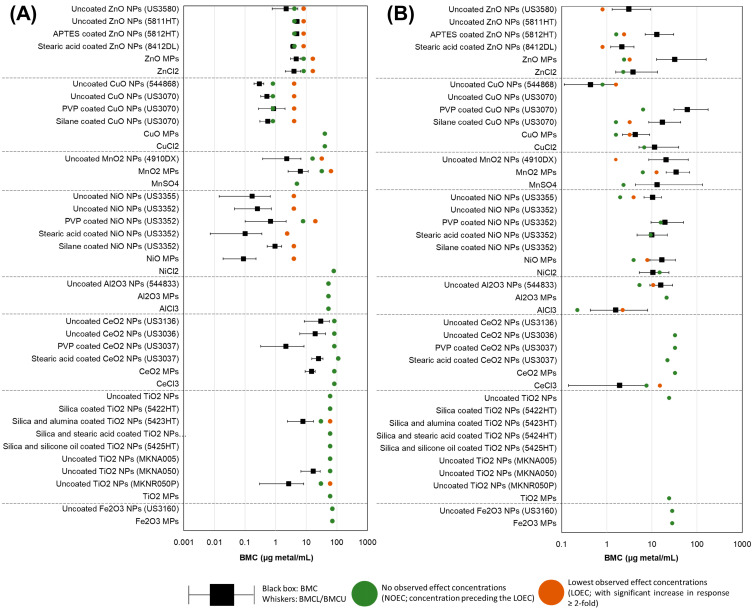
BMC analysis of (**A**) 4 h % DNA in tail, and (**B**) 40 h % MN induction endpoints for MONPs, MOMPs, and dissolved metal exposures. Concentration is expressed in terms of the mass volume of constituent metal. Black dashed lines denote the separation between MO types. BMCL: Lower 95% confidence interval of the BMC. BMCU: Upper 95% confidence interval of the BMC.

**Figure 12 nanomaterials-14-00743-f012:**
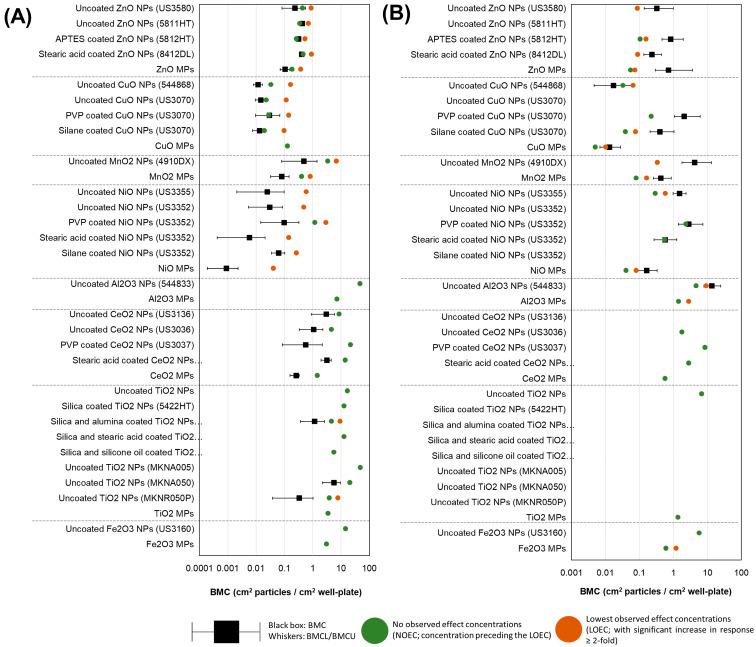
BMC analysis of (**A**) 4 h % DNA in tail, and (**B**) 40 h % MN induction endpoints for MONPs and MOMPs exposures. Concentration is expressed in terms of the SSA of the particles per surface area of the well plate. Black dashed lines denote the separation between MO types. BMCL: Lower 95% confidence interval of the BMC. BMCU: Upper 95% confidence interval of the BMC.

**Figure 13 nanomaterials-14-00743-f013:**
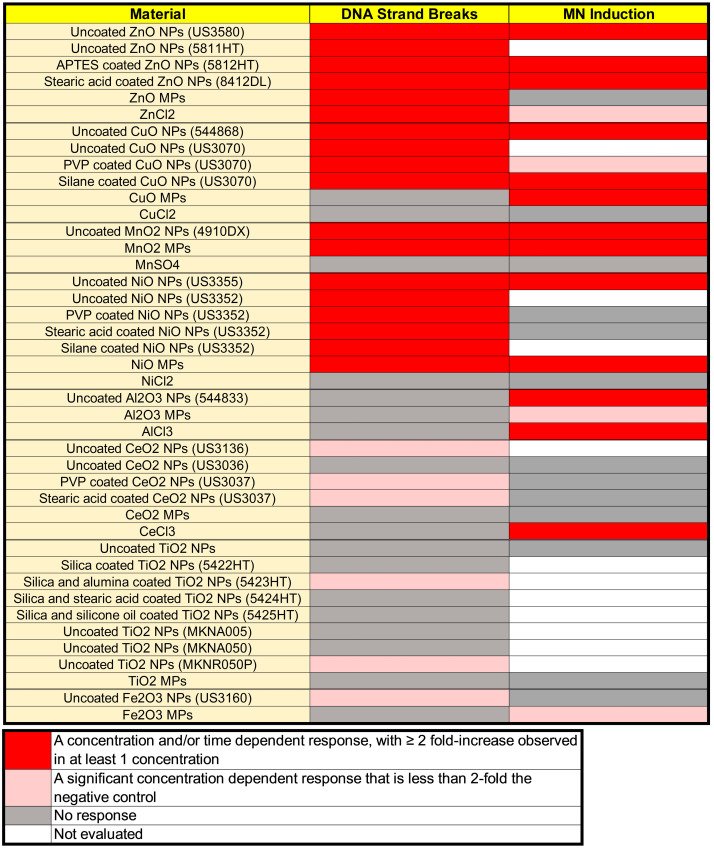
Summary of the results of 4 h comet and 40 h MN assays for all MO and dissolved metal analogs.

**Table 1 nanomaterials-14-00743-t001:** Characteristics of MONPs as provided by the manufacturer.

MONPs	Manufacturer	Coating	Purity (%)	Primary Size (nm)	SSA (m^2^/g)
ZnO US3580	US Research Nanomaterials, Inc., Houston, TX, USA	Uncoated	99+	35–45	65
ZnO 5811HT	Nanostructured and Amorphous Materials, Inc., Katy, TX, USA	Uncoated	99+	30	>15
ZnO 5812HT	Nanostructured and Amorphous Materials, Inc., Katy, TX, USA	APTES-coated	98+	30	>8
ZnO 8412DL	SkySpring Nanomaterials, Inc., Houston, TX, USA	Stearic acid-treated	99	10–30	>60
CuO 544868	Sigma–Aldrich, Oakville, ON, Canada	Uncoated	97.8	<50	29
CuO US3070	US Research Nanomaterials, Inc., Houston, TX, USA	Uncoated	99	40	~45
CuO US3070	US Research Nanomaterials, Inc., Houston, TX, USA	PVP-coated	99	40	~45
CuO US3070	US Research Nanomaterials, Inc., Houston, TX, USA	Silane-coated	99	40	~45
MnO_2_ 4910DX *	SkySpring Nanomaterials, Inc., Houston, TX, USA	Uncoated	98+	40–60	~13.5
NiO US3355	US Research Nanomaterials, Inc., Houston, TX, USA	Uncoated	99.5+	15–35	50–100
NiO US3352	US Research Nanomaterials, Inc., Houston, TX, USA	Uncoated	99.98	18	50–100
NiO US3352	US Research Nanomaterials, Inc., Houston, TX, USA	PVP-coated	99.98	18	50–100
NiO US3352	US Research Nanomaterials, Inc., Houston, TX, USA	Stearic acid-coated	99.98	18	50–100
NiO US3352	US Research Nanomaterials, Inc., Houston, TX, USA	Silane-coated	99.98	18	50–100
Al_2_O_3_ 544833	Sigma–Aldrich, Oakville, ON, Canada	Uncoated	-	<50	>40
CeO_2_ US3136	US Research Nanomaterials, Inc., Houston, TX, USA	Uncoated	99.97	50	30–35
CeO_2_ US3036	US Research Nanomaterials, Inc., Houston, TX, USA	Uncoated	99.97	10–30	30–50
CeO_2_ US3037	US Research Nanomaterials, Inc., Houston, TX, USA	PVP-coated	99.99	10	35–70
CeO_2_ US3037	US Research Nanomaterials, Inc., Houston, TX, USA	Stearic acid-coated	99.99	10	35–70
TiO_2_ NIST ^#^	National Institute of Standards and Technology (NIST,1989), Gaithersburg, MD, USA	Uncoated	99.4 ^a^	19-Anatase (76%) 37-Rutile (24%)	55.55
Fe_2_O_3_ US3160	US Research Nanomaterials, Inc., Houston, TX, USA	Uncoated	99.5+	30	20–60

SSA: Specific surface area. APTES: (3-Aminopropyl) triethoxysilane. PVP: Polyvinylpyrrolidone. * 4910DX is sold as Mn_2_O_3_, but it was identified as MnO_2_ by X-ray diffraction in Avramescu et al. [39]. ^#^ TiO_2_ NIST NPs will be referred to as “uncoated TiO_2_ NPs”. - Not provided by the manufacturer. ^a^ Avramescu et al. [41].

**Table 2 nanomaterials-14-00743-t002:** Characteristics of MOMPs as provided by the manufacturer.

MOMPs	Manufacturer	Purity (%)	Primary Size (µm)	SSA (m^2^/g)
ZnO US1003M	US Research Nanomaterials, Inc., Houston, TX, USA	99.9+	1	2–15.8
CuO US1140M	US Research Nanomaterials, Inc., Houston, TX, USA	99.5	5	4–6
MnO_2_ 4930DX	SkySpring Nanomaterials, Inc., Houston, TX, USA	99+	~5 (50%), <10 (90%)	-
NiO US1014M	US Research Nanomaterials, Inc., Houston, TX, USA	99+ ^a^	5	5–20
Al_2_O_3_ 1331DL	SkySpring Nanomaterials, Inc., Houston, TX, USA	99.9	0.4–1.5	~110
CeO_2_ 2118CG	SkySpring Nanomaterials, Inc., Houston, TX, USA	99.9	5–20	5–8
TiO_2_ US1017M	US Research Nanomaterials, Inc., Houston, Tx, USA	99.9+	1.5 (Anatase) 1.5 (Rutile)	5–8
Fe_2_O_3_ US1139M	US Research Nanomaterials, Inc., Houston, TX, USA	99	5	-

SSA: Specific surface area. - Not provided by the manufacturer. ^a^ Avramescu et al. [41].

**Table 3 nanomaterials-14-00743-t003:** Characterization of MONPs as determined by TEM, DLS, and ELS.

MONPs	Primary Size ^d^ (nm)	Aspect Ratio ^e^	Cell Culture Medium ^f^		
	Length Width				
			HD (nm)	PDI	ZP (mv)
Uncoated ZnO (US3580)	(23.9 ± 7.2) ^a^ (19.4 ± 5.5) ^a^	1.23 ± 0.17 ^a^	323 ± 125 ^a^	0.56 ± 0.12 ^a^	−9.21 ± 1.15
Uncoated ZnO (5811HT)	(44.06 ± 19.4) (33.88 ± 13.2)	1.31 ± 0.32	290 ± 13.5	0.27 ± 0.05	−9.05 ± 0.65
APTES-coated ZnO (5812HT)	(53.51 ± 38.5) (40.4 ± 25.6)	1.31 ± 0.30	322 ± 24.8	0.29 ± 0.05	−10.8 ± 0.60
Stearic acid-treated ZnO (8412DL)	(33.69 ± 16.57) (25.06 ± 9.56)	1.35 ± 0.36	361 ± 65.9	0.42 ± 0.04	−9.00 ± 0.96
Uncoated CuO (544868)	(64.8 ± 47.0) ^b,c^ (45.9 ± 28.0) ^b,c^	1.39 ± 0.39 ^b,c^	396 ± 34.9	0.35 ± 0.05	−11.7 ± 0.73
Uncoated CuO (US3070)	(75.97 ± 42.87) (58.06 ± 34.36)	1.35 ± 0.32	265 ± 8.79	0.26 ± 0.03	−11.7 ± 0.73
PVP-coated CuO (US3070)	(63.07 ± 34.05) (48.07 ± 24.7)	1.32 ± 0.28	272 ± 12.6	0.29 ± 0.04	−11.3 ± 0.57
Silane-coated CuO (US3070)	(84.21 ± 55.46) (62.35 ± 36.28)	1.35 ± 0.41	494 ± 49.1	0.34 ± 0.11	−14.9 ± 0.75
Uncoated MnO_2_ (4910DX)	(36.06 ± 34.88) (13.12 ± 6.33)	2.80 ± 1.97	151 ± 2.42	0.19 ± 0.03	−12.3 ± 0.87
Uncoated NiO (US3355)	(27.29 ± 10.26) ^c^ (21.84 ± 7.90) ^c^	1.25 ± 0.20 ^c^	229 ± 18.5	0.30 ± 0.05	−12.9 ± 1.03
Uncoated NiO (US3352)	(25.56 ± 13.16) (20.71 ± 10.08)	1.23 ± 0.24	231 ± 4.11	0.21 ± 0.02	−14.4 ± 1.06
PVP-coated NiO (US3352)	(31.99 ± 14.7) (24.66 ± 11.28)	1.31 ± 0.29	213 ± 4.37	0.20 ± 0.02	−13.6 ± 0.52
Stearic acid-coated NiO (US3352)	(29.59 ± 14.42) (22.81 ± 10.38)	1.30 ± 0.25	285 ± 9.62	0.28 ± 0.05	−16.7 ± 0.82
Silane-coated NiO (US3352)	(29.51 ± 12.46) (23.90 ± 9.98)	1.24 ± 0.19	222 ± 8.49	0.29 ± 0.04	−13.6 ± 0.83
Uncoated Al_2_O_3_ (544833)	(23.92 ± 11.84) ^c^ (10.68 ± 6.85) ^c^	2.63 ± 1.40 ^c^	385 ± 48.6	0.37 ± 0.05	−12.4 ± 0.81
Uncoated CeO_2_ (US3136)	(39.78 ± 14.55) (31.25 ± 9.66)	1.27 ± 0.22	345 ± 6.72	0.27 ± 0.03	−14.0 ± 1.12
Uncoated CeO_2_ (US3036)	(12.05 ± 3.42) (9.90 ± 2.73)	1.22 ± 0.20	357 ± 15.2	0.32 ± 0.05	−13.9 ± 1.91
PVP-coated CeO_2_ (US3037)	(14.12 ± 4.54) (10.92 ± 3.42)	1.29 ± 0.22	383 ± 20.4	0.32 ± 0.05	−11.9 ±0.99
Stearic acid-coated CeO_2_ (US3037)	(10.61 ± 2.71) (8.23 ± 2.05)	1.30 ± 0.20	405 ± 23.4	0.40 ± 0.05	−12.2 ± 0.68
Uncoated TiO_2_ NPs	(26.8 ± 8.9) ^a,c^ (20.8 ± 6.8) ^a,c^	1.30 ± 0.26 ^c^	421 ± 31 ^c^	0.23 ± 0.06 ^c^	−11.5 ± 0.2 ^c^
Uncoated Fe_2_O_3_ (US3160)	(30.95 ± 13.12) (24.99 ± 11.24)	1.26 ± 0.24	193 ± 12.9	0.28 ± 0.03	−11.4 ± 0.86

^a^ Characterization described by Boyadzhiev et al. [34]. ^b^ Characterization described by Boyadzhiev et al. [49]. ^c^ Previously reported in Boyadzhiev et al. [54]. ^d^ Primary size determined by TEM. ^e^ Aspect ratio: Length/Width. ^f^ DMEM/F12 cell culture medium without phenol red + 2% FBS + 1 ng/mL EGF. HD, PDI, and ZP were determined by DLS and ELS (characterization in dH_2_O can be found in Appendix A).

## Data Availability

The data presented in this study are available on request from the corresponding author.

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
