# Peer review of "A Systematic Genotoxicity Assessment of a Suite of Metal Oxide Nanoparticles Reveals Their DNA Damaging and Clastogenic Potential"

_nanomaterials, 2024, doi:10.3390/nano14090743_

Round 1

Reviewer 1 Report

Comments and Suggestions for Authors

This manuscript describes the in vitro genotoxic potential assessment of ZnO, CuO, MnO2, NiO, Al2O3, CeO2, TiO2, Fe2O3, NPs, MPs, and dissolved metal analogs using high-throughput CometChip and MicroFlow assays. Although the study was carefully constructed and the results presented were convincing, the length of the manuscript seems to have detracted from its significance. Authors should keep their manuscripts as concise as possible. Although it goes against the grain to shorten the manuscript, the authors should outline how to read Figures 8, 9, 11, and 12 for better understanding. Also, regarding Figure 13, it is easier to understand if the meaning of each is shown next to each colored column on the side or above the figure, rather than summarizing them in a legend. In addition, there are many points which the authors should address as follows;

1.     Why did the authors focus on metal oxide nanoparticles rather than metal nanoparticles?

2.     In lines 137 and 140, please give an easy-to-understand explanation about the sudden appearance of MN.

3.     In the last paragraph of the "Introduction", please also add the types of coatings that are an important pillar of this research and the reasons why the authors are investigating them.

4.     In Section 2.6, please add the reason why EF1 cells were selected.

5.     In Section 2.9, please add a brief explanation of why MOMPs were added to the experiment. Why did the authors choose student t-test instead of ANOVA? Did they only compare between two groups?

6.     In Table 3, what did it mean to know the dynamics of MONP in the culture medium? Isn't the dynamics in the environment better?

7.     What was the most important reason for the difference in DNA damage ability of each target substance?

8.     Figure 6 showed significant differences in Ce, but were these differences physiologically meaningful?

Reviewer 2 Report

Comments and Suggestions for Authors

This study utilized micronucleus assays to examine DNA and chromosomal damage in mouse lung epithelial cells. The research investigated the effects induced by nano and bulk sizes of zinc oxide, copper oxide, manganese oxide, nickel oxide, aluminum oxide, cerium oxide, titanium dioxide, and iron oxide.

This is an interesting work which can be published in nanomaterials after some revision

All authors of this article, except the corresponding one, share the same address. Therefore, the address should not be repeated six times; instead, each author can be indexed to the same address.

Abstract

Consistency in the use of abbreviations is crucial for clarity. The text alternates between "MONPs" and "MONP" when referring to metal oxide nanoparticles.

The statement "Ionic forms of MONPs were included, where available" lacks specificity regarding the inclusion or exclusion of specific ionic forms. Providing specific information would improve clarity and understanding, or alternatively, omitting this statement altogether could be considered.

 “Surface coating of MONPs inconsistently affected genotoxicity” It's unclear what "genotoxicity" refers to in the context of surface coating inconsistently affecting it. clarify

Introduction

Engineered Nanomaterials (ENMs): The text defines ENMs as materials manufactured at or within the nanoscale (1 to 100 nanometres inclusive) or having internal or surface structures in the nanoscale. However, it does not clearly specify what is meant by "internal or surface structures," which could lead to confusion or misinterpretation.

The occupational exposure limits  for MONPs are suggested to may not be protective enough based on studies showing signs of systemic toxicity at exposure levels of 1 mg/m3, However, the relevance of this finding to current occupational exposure limits is unclear, as the text does not provide specific information on existing regulations or guidelines for MONP exposure.

Various toxicity endpoints, including inflammation, oxidative stress, altered expression of antioxidant enzymes, and neurotoxicity are mention, however, it does not clearly define these endpoints or specify how they were assessed in the referenced studies.

"stearic acid-coated NiO2 NPs,"  NiO2,  is not a standard chemical formula for nickel oxide. The correct chemical formula for nickel oxide is "NiO." Ensuring the use of precise and consistent terminology.

Experimental

Cell viability assay to determine the experimental conditions for the comet assay. The text mentions that cells were plated at 65,000 cells/cm² in a 6-well plate. However, the standard practice for Trypan Blue exclusion analysis typically involves counting cells in suspension rather than plating them at a specific density.

The formula provided for calculating cell viability (% cell viability = [Live cells/Total cell count] * 100) seems inconsistent with the subsequent calculation of relative survival. Typically, cell viability is expressed as the percentage of live cells relative to the total number of cells counted, as indicated by Trypan Blue staining. However, the formula for relative survival (% relative survival = [Number of live cells/cm² of samples] / [number of live cells/cm² of the control] * 100) suggests a comparison between treated samples and a control group rather than an absolute measure of viability.

Statistical Analysis: The text mentions the use of a Kruskal-Wallis test with Dunnett’s post-hoc test for data analysis. While the Kruskal-Wallis test is suitable for comparing multiple groups when the data are not normally distributed, Dunnett’s post-hoc test is typically used to compare treatment groups against a single control group, not against each other. A more appropriate post-hoc test for multiple comparisons would be the Dunn-Bonferroni test or Tukey's test.

Cell viability and relative survival

The uncoated MnO2 NPs and Fe2O3 NPs, along with their microparticle (MP) forms, were evaluated after 24 hours of exposure because their NP and MP forms did not induce a decrease in cell viability. However, this statement seems contradictory. If the NP forms did not induce cytotoxicity, it raises questions about the necessity of evaluating their MP forms.

Evaluation of ZnO Variants: The text mentions that the highest concentration evaluated for ZnO variants was set to 20 μg/mL due to cytotoxic effects observed at higher concentrations. However, no specific details are provided regarding the concentrations tested for other variants or analogues. Providing this information would enhance the clarity and reproducibility of the study.

Consistency in Reporting Results: While the text mentions that percent cell viability did not alter after exposure to metal oxide variants and dissolved metal analogues, the specific results for each variant or analogue are not provided in the excerpt. Including these results would allow readers to assess the impact of different materials on cell viability accurately.

Statistical Analysis: The text does not mention whether statistical analysis was performed to assess the significance of differences in cell viability between treatment groups and controls. Including information about statistical tests and significance levels would strengthen the scientific rigor of the study.

While the text mentions changes in % relative survival for various materials at different concentrations and exposure durations, it does not specify whether these changes were statistically significant. Providing information about statistical analysis and significance levels would allow readers to assess the reliability of the observed effects.

Lack of Complete Reporting: While the text provides some results for % relative survival at different concentrations and exposure durations, it does not include comprehensive data for all materials tested. Providing complete data for all materials would allow for a more thorough evaluation of the experimental findings.

Comments on the Quality of English Language

Minor editing of English language is required

Reviewer 3 Report

Comments and Suggestions for Authors This manuscript, which is both interesting and well-written, deals with the genotoxicity testing of certain metal oxide nano- and microparticles. The paper is scientifically sound and is certainly worthy of publication in Nanomaterials. I have only a few minor remarks or concerns.
  1. The title of the paper is somewhat unspecific and seems more suited to a review than a research paper. Moreover, it’s clear that not all metal oxide NPs were investigated in this research.

  2. The introduction is overly lengthy and resembles a review more than a typical introduction to the problem. It could be easily shortened by moving some parts into the Discussion section.

  3. Since FE1 cells were used, why weren’t point mutations evaluated in parallel?

  4. Does log transformation work for the normalization of comet data? Have you tested this?

  5. In the MicroFlow MN Assay, the usual treatment time is 24 hours. Why did you use a 40-hour treatment time?

  6. The data generated by the MicroFlow MN Assay is quite specific and differs from data generated by the classical MN assay. In particular, although many papers using the MicroFlow MN assay indicate that they are measuring MN frequency (expressed as %MN), this is not correct - one can calculate a percentage from the same, identical events. Therefore, the index %MN makes no sense at all. That’s why other papers use a more appropriate index - the fold-increase in MN frequency. I highly recommend that the authors of the present paper use only the expression “fold-increase”.

  7. The statistical analysis of the fold-increase is not straightforward, and common statistical methods, such as the t-test, are not the best solution. Rules-based decisions are typically used (see, for example, Wild et al., Environmental and Molecular Mutagenesis 58:662-677, 2017).

  8. The MicroFlow MN Assay has relatively low sensitivity and specificity as a test for genotoxicity. This should be acknowledged somewhere in the text.

  9. In Figures 8 and 9, what does the dashed line represent - 40% of relative survival or a 2-fold change in MN frequency? Please specify.

  10. There is little point in indicating different levels of significance (for example, *p<0.05, **p<0.01,***p<0.001) - it adds no additional information.

  11. Not all figures are clearly readable. For example, the positive control values are high relative to the experimental data in Figures 1-7. It would be helpful to have a slightly higher Y-axis. In the Supplementary Materials, in Figures S9-S17, the lower values on the Y-axis could be cut in order to zoom in on the upper parts of the columns (let’s say, the lowest value on the Y-axis could be 40 or 60, and the highest - 140).

Round 2

Reviewer 1 Report

Comments and Suggestions for Authors

The reviewer thinks that the authors were probably misled by the expression "Although it goes against the grain to shorten the manuscript," but there is no response to the reviewer's opinion "Authors should keep their manuscripts as concise as possible.".

Honestly, this manuscript seems too long. However, if the authors insist that all of these are necessary, please add a valid reason.

5. Regarding statistical analysis, the authors explained that one-way ANOVA requires a normal distribution, but the Student's T-test similarly assumes that both groups are normally distributed. Masu. What do the authors think about this? Also, in this section, Student’s T-test is Student’s T-test without any correspondence, right?

7. The authors mentioned that multiple factors contribute to the ability to damage DNA, but what is the proof that all the results shown in this study were the cause rather than the effect?

The experiments were well conducted. After appropriate revisions, the manuscript can be accepted.
